# Neural Spectral Marked Point Processes

**Shixiang Zhu**[†]**, Haoyun Wang**[†]**, Zheng Dong**[†]**, Xiuyuan Cheng**[⋆]**, Yao Xie**[†]
† Georgia Institute of Technology
⋆ Duke University
`yao.xie@isye.gatech.edu`

## Abstract

Self- and mutually-exciting point processes are popular models in machine learning and statistics for dependent discrete event data. To date, most existing models assume stationary kernels (including the classical Hawkes processes) and simple parametric models. Modern applications with complex event data require more general point process models that can incorporate contextual information of the events, called marks, besides the temporal and location information. Moreover, such applications often require non-stationary models to capture more complex spatio-temporal dependence. To tackle these challenges, a key question is to devise a versatile influence kernel in the point process model. In this paper, we introduce a novel and general neural network-based non-stationary influence kernel with high expressiveness for handling complex discrete events data while providing theoretical performance guarantees. We demonstrate the superior performance of our proposed method compared with the state-of-the-art on synthetic and real data.

## 1 Introduction

Event sequence data are ubiquitous in our daily life, ranging from traffic incidents, 911 calls, social media posts, earthquake catalog data, and COVID-19 data (see, e.g., Bertozzi et al. (2020)). Such data consist of a sequence of events indicating when and where each event occurred, with additional descriptive information (called *marks*) about the event (such as category, volume, or free-text). The distribution of events is of scientific and practical interest, both for prediction purposes and for inferring events' underlying generative mechanism.

A popular framework for modeling events is point processes (Daley & Vere-Jones, 2008), which can be continuous over time and the space of marks. An important aspect of this model is capturing the event's triggering effect on its subsequent events. Since the distribution of point processes is completely specified by the conditional intensity function (the occurrence rate of events conditioning on the history), such triggering effect has been captured by an influence kernel function embedded in the conditional intensity. In statistical literature, the kernel function usually assumes a parametric form. For example, the original work by Hawkes (Hawkes, 1971) considers an exponential decaying influence function over time, and the seminal work (Ogata, 1998) introduces epidemic-type aftershock sequence (ETAS) model, which considers an influence function that exponentially decays over space and time. With the increasing complexity of modern applications, there has been much recent effort in developing recurrent neural network (RNN)-based point processes, leveraging the rich representation power of RNNs (Du et al., 2016; Mei & Eisner, 2017; Xiao et al., 2017b).

However, there are several limitations of existing RNN-based models. First, such models typically do not consider the kernel function (Du et al., 2016; Li et al., 2018; Mei & Eisner, 2017; Upadhyay et al., 2018; Xiao et al., 2017a;b); thus, the RNN approach does not enjoy the interpretability of the kernel function based models. Second, the popular RNN models such as Long Short-Term Memory (LSTM) (Hochreiter & Schmidhuber, 1997) still implicitly discounts the influence of events over time (due to their recursive structure) (Vaswani et al., 2017; Zhu et al., 2021d). Such assumptions may not hold in many real-world applications. Take the earthquake catalog as an example, which is a typical type of discrete event data; most aftershocks occur along the fault plane or other faults within the volume affected by the mainshock's strain (Zhu et al., 2021b). This means that different regions may be correlated to their surrounding area differently according to their geological structure, which creates a complex non-stationary spatial profile that we would like to capture through the model. Third, a majority of the existing works mainly focus on one-dimensional temporal point

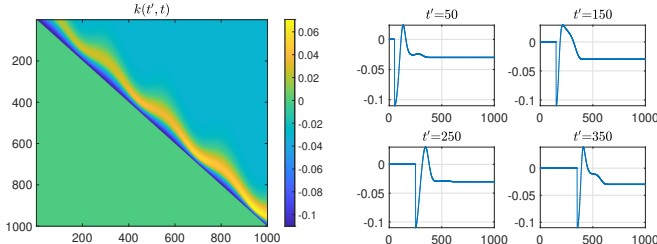

Figure 1: An example of non-stationary influence kernel $k(t', t)$ of event time $t'$ and future time $t > t'$.

processes. Although there are works on marked point processes (Du et al., 2016; Mei & Eisner, 2017; Reinhart, 2018), they are primarily based on simplifying assumptions that the marks are conditionally independent of the event's time and location, which is equivalent to assuming the kernel is separable; these assumptions may fail to capture some complex non-stationary, time- and location-dependent triggering effects for various types of events, as observed for many real-world applications (see, e.g., (Bertozzi et al., 2020)).

**Contribution.** In this paper, we present a novel general non-stationary point process model based on neural networks, referred to as the neural spectral marked point process (NSMPP). The key component is a new powerful representation of the kernel function using neural networks, which enables us to go beyond stationarity (and thus go beyond standard Hawkes processes) and has the capacity to model high-dimensional marks. Figure 1 gives an example of non-stationary influence kernel that measures the influence of the past events to the future time $t$. The premise of the model design is that the conditional intensity function uniquely specifies the distribution of the point process, and the most important component in the intensity function is the influence kernel. In summary, the novelty of our approach includes the following:

- The kernel function is represented by a spectral decomposition of the influence kernel with a finite-rank truncation in practice. Such a kernel representation will enable us to capture the most general non-stationary process as well as high-dimensional marks. The model also allows the distribution of marks to depend on time, which is drastically different from the separable kernels considered in the existing literature (Reinhart, 2018).
- The spectral decomposition of asymmetric influence kernel consists of a sum of the product of feature maps, which can be parameterized by neural networks. This enable us to harvest the powerful expressiveness and scalability to high-dimensional input of neural networks for complicated tasks involving discrete events data.
- We establish theoretical guarantees of the maximum likelihood estimate for the true kernel function based on functional variational analysis and finite-dimensional asymptotic analysis, which shed light on theoretical understanding of neural network-based kernel functions.
- Using synthetic and real data (seismic and police data), we demonstrate the superior performance of our proposed method in complex situations; the performance gain is particularly outstanding for cases involving non-stationary point processes.

**Related work.** Seminal works in point processes modeling (Ogata, 1988; 1998) assume parametric forms of the intensity functions. Such methods enjoy good interpretability and are efficient to estimate. However, classical parametric models are not expressive enough to capture the events' dynamics in modern applications.

Recent research interests aim to improve the expressive power of point process models, where a recurrent neural networks (RNNs)-based structure is introduced to represent the conditional intensity function (Du et al., 2016; Mei & Eisner, 2017; Xiao et al., 2017b). However, most of these works either explicitly or implicitly specify the inter-event dependence in a limited form with restrained representative power. For example, Du et al. (2016) expresses the influence of two consecutive events in a form of $\exp\{w(t_{i+1} - t_i)\}$, which is an exponential function with respect to the length of the time interval $t_{i+1} - t_i$ with weights $w$; Mei & Eisner (2017) enhances the expressiveness of the model and represents the entire history using the hidden state of an LSTM, which still implicitly assumes the influence of the history decays over time due to the recurrent structure of LSTM.

Another line of research uses neural networks to directly model dependence of sequential events without specifying the conditional intensity function explicitly (Li et al., 2018; Xiao et al., 2017a). Some studies consider non-stationary influence kernel using neural networks in spatio-temporal

point processes (Zhu et al., 2021b;a). Recent work (Omi et al., 2019) also uses a neural network to parameterize the hazard function, the derivative of which gives the conditional intensity function. However, the above approaches either capture the temporal dependence or assume the temporal and mark dependence are separable rather than jointly accounting for marked-temporal dependence.

Recently, attention models have become popular in computer vision and sequential data modeling (Britz et al., 2017; Luong et al., 2015; Vaswani et al., 2017). This motivates works including Zhang et al. (2019); Zhu et al. (2021c;d); Zuo et al. (2020) to model the conditional intensity of point processes using the attention mechanism and characterize the inter-event dependence by a score function. The attention mechanism has proven to be more flexible in capturing long-range dependence regardless of how far apart two events are separated and greatly enhances the performance in practice. However, the main limitation of Zhang et al. (2019); Zuo et al. (2020) is that they rely on a conventional score function – dot-product between linear mappings of events, which is still limited in representing non-linear dependence between events for some applications. Zhu et al. (2019; 2021c;d) used a more flexible and general Fourier kernel as a substitution for the dot-product score; however, the expressive power of the proposed Fourier kernel is still limited, and the spectrum of the Fourier basis is represented by a generative neural network, which is difficult to learn in some cases (Arjovsky & Bottou, 2017).

There are also works considering point processes with non-stationary intensities. Chen & Hall (2013) proposed time-varying background intensity for point process, while we focus on the non-stationary triggering kernel depicting complex events dependency. Remes et al. (2017; 2018) studied non-stationary kernels combined with Gaussian processes, assuming specific structures of the kernels in the Fourier domain. Such kernels are more restricted than ours since the nature of Gaussian processes requires that the kernel is positive semidefinite.

## 2 METHOD

### 2.1 BACKGROUND: MARKED TEMPORAL POINT PROCESS

Marked temporal point processes (MTPPs) (Reinhart, 2018) consist of a sequence of events over time. Each event is associated with a (possibly multi-dimensional) *mark* that contains detailed information of the event, such as location, nodal information (if the observations are over networks, such as sensor or social networks) categorical data, and contextual information (such as token, image, and text descriptions). Let $T > 0$ be a fixed time-horizon, and $\mathcal{M} \subseteq \mathbb{R}^d$ be the space of marks. We denote the space of observation as $\mathcal{X} = [0, T) \times \mathcal{M}$ and a data point in the discrete event sequence as

$$x = (t, m), \quad t \in [0, T), \quad m \in \mathcal{M}, \tag{1}$$

where $t$ is the event time and $m$ represents the mark. Let $N_t$ be the number of events up to time $t < T$ (which is random), and $\mathcal{H}_t := \{x_1, x_2, \dots, x_{N_t}\}$ denote historical events. Let $\mathbb{N}$ be the counting measure on $\mathcal{X}$, i.e., for any measurable $S \subseteq \mathcal{X}$, $\mathbb{N}(S) = |\mathcal{H}_T \cap S|$. For any function $f : \mathcal{X} \to \mathbb{R}$, the integral with respect to the counting measure is defined as

$$\int_S f(x) d\mathbb{N}(x) = \sum_{x_i \in \mathcal{H}_T \cap S} f(x_i).$$

The events' distribution in MTPPs can be characterized via the *conditional intensity function* $\lambda(x)$, which is defined to be the conditional probability of observing an event in the marked temporal space $\mathcal{X}$ given the events' history $\mathcal{H}_{t(x)}$, that is,

$$\mathbb{E}\left(d\mathbb{N}(x)|\mathcal{H}_{t(x)}\right) = \lambda(x)dx. \tag{2}$$

Above, $t(x)$ extracts the occurrence time of event $x$, and we omit the dependence on $\mathcal{H}_{t(x)}$ in the notation of $\lambda(x)$ for simplicity.

As self- and mutual-exciting point processes, Hawkes processes (Hawkes, 1971) have been widely used to capture the mutual excitation dynamics among temporal events. The model assumes that influences from past events are linearly additive towards the current event. The conditional intensity function for a self-exciting point process takes the form of

$$\lambda[k](x) = \mu + \sum_{x' \in \mathcal{H}_{t(x)}} k(x', x), \tag{3}$$

where $\mu > 0$ stands for the background intensity, and the so-called "influence kernel" $k : \mathcal{X} \times \mathcal{X} \to \mathbb{R}$ is crucial in capturing the influence of past events on the likelihood of event occurrence at the current time. Here we use the notation $[k]$ to stress the dependence of the conditional intensity function on the kernel function $k(x', x)$. Written in the form of the integral over counting measure, we have that

$$\lambda[k](x) = \mu + \int_{x' \in \mathcal{X}_{t(x)}} k(x', x) d\mathbb{N}(x'), \qquad (4)$$

where $\mathcal{X}_t$ is the subset of $\mathcal{X}$ with the first component smaller than $t$.

The most commonly made assumption in the literature is that the process is stationary, where the influence of the past events is shift-invariant, such that $k(x', x) = f(x - x')$ for a influence function $f : \mathbb{R}^d \to \mathbb{R}^+$; a common influence function in one-dimensional cases is $f(t) = \alpha \exp\{-\beta t\}$, where $\beta$ controls the decay rate and $\alpha > 0$ controls the magnitude of the influence of an event. The current work aims at going beyond stationary point processes, which enables us to better capture the heterogeneity in the events' influence across the spatial-temporal space, which naturally arises in many applications.

## 2.2 Neural spectral representation for influence kernel

We propose to represent more general non-stationary influence kernels in the conditional intensity function $\lambda(x)$ specified in (4).

**Kernel representation.** The main idea of the proposed model is to represent the influence kernel $k$ using a general finite-rank decomposition

$$k(x', x) = \sum_{r=1}^{R} \nu_r \psi_r(x') \phi_r(x), \quad \nu_r \geq 0, \qquad (5)$$

where

$$\psi_r : \mathcal{X} \to \mathbb{R}, \quad \phi_r : \mathcal{X} \to \mathbb{R}, \quad r = 1, \cdots, R,$$

are two sets of *feature functions* in some smooth functional space $\mathcal{F} \subset C^0(\mathcal{X})$, and $\nu_r$ is the corresponding weight - or "spectrum". This representation is motivated by the spectral decomposition of a general kernel function. While functional spectral decomposition is usually infinitely dimensional, for practical considerations, we truncate the "spectrum" and only consider a finite rank representation. Note that while we view (5) as similar to a spectral decomposition, it can be better understood as feature maps in kernel representation, and, particularly, we do not need the feature functions $\psi_r$ and $\phi_r$ to be orthogonal.

The decomposition (5) represents the kernel function using three parts: two sets of (normalized) feature functions and the energy spectrum—the spectrum $\nu_r$ plays the role of weights to combine the feature maps. In the learning process, we can train the feature functions (typically neural networks) and the weights separately; since learning the normalized feature maps tend to be more numerically stable. The proposed form of kernel is not necessarily positive semi-definite, and even not symmetric.

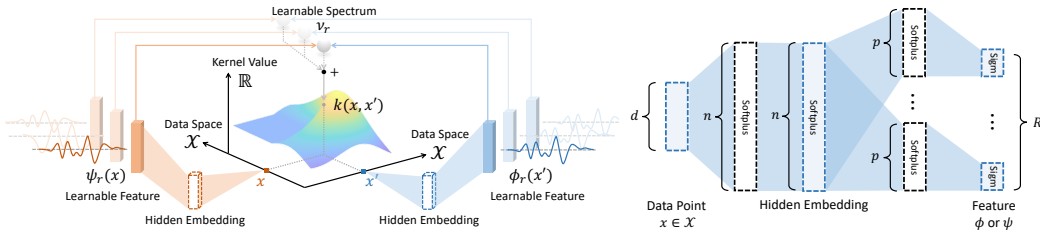

(a) Kernel architecture          (b) Network structure

Figure 2: (a) The architecture of the proposed non-stationary neural spectral kernel. A hidden embedding will first summarize the data point; then, the embedding will be mapped to different features via multi-branch neural networks. The kernel value is calculated by summing up the products between two sets of learned features, $\{\psi_r\}$ and $\{\phi_r\}$, weighted by the spectrum $\{\nu_r\}$. The spectrum, feature functions, and hidden embeddings are jointly learned from data. (b) The structure of the multi-branch neural network. There are two shared layers with $n$ nodes per layer that generate the hidden embedding; $d$ denotes the dimension of the input data point; $p$ denotes the dimension of the middle layer that generates the feature. Further specifications of neural networks will be provided in the experiments in Section 4.

Note that the spectral representation can allow for a completely general kernel that can be used for non-stationary processes (since our influence function does not impose a shift-invariant structure). Moreover, the spectral representation allows us to go beyond monotone decay or parametric form as commonly assumed in the prior literature.

**Neural network feature function.** As one of the most salient features of our method, feature functions $\{\phi_r, \psi_r\}$ are represented using neural networks, leveraging their known universal approximation power. First, the input data point $x \in \mathcal{X}$ will be projected to a hidden embedding space via a multi-layer shared network, aiming to extract key information of the input data. Here we have adopted Softplus non-linear activation in this feature extraction sub-network, while other options may be possible. Next, the hidden embedding will be mapped to the different features $\{\phi_r(x)\}$ through $R$ branched sub-networks. To ensure the output feature is constrained in a bounded space, we choose the scaled sigmoidal function as the activation of the output layer, i.e., $f(x) = s/(1 + \exp(-x))$, where $s$ is a constant to enable rescaling of the output to a proper range (in our setting, we set $s$ to be 100). The overall architecture of our kernel formulation and the structure of the multi-branch neural network are summarized in Figure 2 (a) and (b), respectively. Further specifications of neural networks will be provided in the experiments in Section 4.

### 2.3 MAXIMUM LIKELIHOOD MODEL RECOVERY

An essential task in learning the neural point process model is to estimate the influence kernel function for the point process. The reason is two-fold: First, the kernel function is the most important component in representing the point process. Second, in practice, the influence kernel offers clear interpretations, such as "how events at a particular time and location will influence future events at a given time and location." Such interpretation is essential for predicting using event data– one of the main applications for point process models.

To estimate the influence kernel function for point process models, we consider a popular approach through maximum likelihood estimation (MLE). Formally, the optimal kernel can be found by solving the following optimization problem given $M$ sequences of training event sequences over the time horizon $[0, T]$: $\{x_{i,j}\}$, $i = 1, ..., N_j$, $j = 1, \ldots, M$:

$$\max_{k \in \mathcal{K}} \ell[k] := \frac{1}{M} \sum_{j=1}^{M} \left( \int_{\mathcal{X}} \log \lambda_j[k](x) d\mathbb{N}_j(x) - \int_{\mathcal{X}} \lambda_j[k](x) dx \right), \tag{6}$$

where $\lambda_j$ and $\mathbb{N}_j$ denote the conditional intensity and counting measure associated with the $j$-th trajectory, respectively, and $\mathcal{K} \subset C^0(\mathcal{X} \times \mathcal{X})$ represents the family of regular kernel functions induced by the feature function family $\mathcal{F}$ and the finite-rank decomposition (5).

**Learning algorithm.** To solve MLE (6) when the kernel function $k$ is parameterized by neural networks, we use a stochastic gradient as summarized in Algorithm 1 (Appendix A). Note that to calculate the log-likelihood function $\ell$ in (6), we need to evaluate an integral (the second term), which does not have a closed-form expression. We approximate the integral numerically by Monte Carlo integration – drawing samples and take the average, as described in Algorithm 2 (Appendix A).

## 3 THEORETICAL GUARANTEES OF MLE

We first consider the MLE as a *functional optimization problem*, since when using neural networks to approximate the kernel function, we are interested in such functional approximation results. We show that the expected log-likelihood reaches its maximum at the true kernel with the second-order derivative bounded away from zero, and thus the *true kernel function is identifiable by solving the MLE problem* under some regularity conditions.

Consider the log-likelihood of point processes over a family of kernel functions $\overline{\mathcal{K}}$ which contains $\mathcal{K}$, the family induced by feature functions in $\mathcal{F}$ and non-negative spectrum $\{\nu_r\}_{r=1}^{R}$. Note that $\overline{\mathcal{K}}$ may go beyond the finite-rank decomposition in (5). Later throughout the theoretical analysis, we omit the spectrum as they can be absorbed into the feature functions. The details are discussed in Remark 3.2.

**Assumption 3.1.** *(A1) The kernel function family $\overline{\mathcal{K}} \subset C^0(\mathcal{X} \times \mathcal{X})$ which is uniformly bounded, and the true kernel $k^* \in \overline{\mathcal{K}}$; (A2) There exist $c_1, c_2$ positive constants, such that for any $k \in \overline{\mathcal{K}}$, a.s. for event data trajectory, $c_1 \leq \lambda[k](x) \leq c_2$, $\forall x \in \mathcal{X}$.*

Note that, apart from (A2), we only need kernel functions to be measurable for theoretical analysis which is guaranteed by (A1). In practice, the proposed neural network parametrization leads to a continuous and smooth (low-rank) kernel function, which induces non-singular $\lambda$.

We have the following lemma, which shows that when the log-likelihood function has a local perturbation around the true kernel function, there is going to be a decrease in the log-likelihood function value in expectation. Note we assume that $k^*$ lies in (or can be well-approximated by) the family of function class $\mathcal{K}$ – thanks to the well-known universal approximation power of neural networks.

**Lemma 3.1** (Local perturbation of likelihood function around the true kernel function)**.** *Under Assumption 3.1, for any $\tilde{k} \in \overline{\mathcal{K}}$ and $\delta k = \tilde{k} - k^*$, we have*

$$\ell[k^*] - \ell[\tilde{k}] \geq \frac{1}{M} \left\{ -\sum_{j=1}^{M} \int_{\mathcal{X}} \delta\lambda_j(x) \left( \frac{d\mathbb{N}_j(x)}{\lambda_j[k^*](x)} - dx \right) + \frac{1}{2c_2^2} \sum_{j=1}^{M} \int_{\mathcal{X}} (\delta\lambda_j(x))^2 d\mathbb{N}_j(x) \right\}, \quad (7)$$

*where*

$$\delta\lambda_j(x) := \int_{\mathcal{X}_{t(x)}} \delta k(x', x) d\mathbb{N}_j(x'). \quad (8)$$

The implication of Lemma 3.1 is the following. Note that in (7), we have nicely decomposed the difference in the likelihood function caused by a small perturbation around the true kernel function, as two terms: the first term in (7) is a martingale integral (since the conditional expectation of $(d\mathbb{N}_j(x)/\lambda_j[k^*](x) - dx)$ is zero, $\forall j$), and the second term is the integral of a quadratic term against the counting measure. The expectation of the first term is zero due to the property of the martingale process. For the second term, per $j$,

$$\int_{\mathcal{X}} (\delta\lambda_j(x))^2 d\mathbb{N}_j(x) \approx \int_{\mathcal{X}} (\delta\lambda_j(x))^2 \lambda_j^*(x) dx \geq c_1 \int_{\mathcal{X}} (\delta\lambda_j(x))^2 dx, \quad (9)$$

and thus perturbation of the intensity function which corresponds to the second (quadratic) term in (7) will be reflected in the decrease of the log-likelihood. Furthermore, we have the identifiability of the kernel itself, as stated in the following theorem.

**Theorem 3.2** (Kernel identifiability using maximum likelihood)**.** *Under Assumption 3.1, the true kernel function $k^*$ is locally identifiable in that $k^*$ is a local minimum solution of maximum likelihood (6) in expectation.*

*Remark* 3.1 (Function identification and neural network parametrization)**.** Theorem 3.2 derives a result which holds for variational perturbation of the kernel function $k$ in the possibly parametric family $\mathcal{K}$ induced by the feature function family $\mathcal{F}$. When $\mathcal{F}$ is the class of functions that a neural network can represent, the perturbation $\delta k$ is induced by the change of network parameters. It is common that in neural network models, parameter identification is difficult to search for (e.g., due to the symmetry of permuting hidden neurons); however, the neural network function identification may still hold, e.g., under the mean-field approximation (Mei et al., 2018). Thus the kernel function identification results in Theorem 3.2 is important when learning kernel functions by neural networks.

*Remark* 3.2 (Finite rank kernel representation)**.** When assuming the parameter representation (5), we represent the true kernel as $k^*(x', x) = \sum_{r=1}^{R} \nu_r \psi_r^*(x') \phi_r^*(x)$. For theoretical analysis purposes, without loss of generality, we can assume $\nu_r = 1$, $r = 1, \cdots, R$, since they can absorbed into the feature functions. Consider a perturbed kernel where the feature functions are $\tilde{\psi}_r = \psi_r + \delta\psi_r$ and $\tilde{\phi}_r = \phi_r + \delta\phi_r$, and remains to satisfy Assumption 3.1. The kernel function variation $\delta k$ in (8) can then be written as $\delta k(x', x) = \sum_{r=1}^{R} (\delta\psi_r(x')\phi_r(x) + \psi_r(x')\delta\phi_r(x) + \delta\psi_r(x')\delta\phi_r(x))$. With a rank-$R$ representation of the kernel and smooth $\psi_r$, $\phi_r$ represented by neural networks, we potentially prevent overfitting that leads to singular kernels, which may be a problem for the over-complete kernel family $\overline{\mathcal{K}}$ as in (A1). Later we show experimentally that the learned kernel is smooth and close to the true kernel in Figure 3 and Figure 4.

We also studied the MLE for parameterized kernel in Appendix D, when the target feature function belongs to a certain parametric function class with a finite number of parameters. This includes, for instance, spline functions, functions represented by Fourier basis, and neural networks, which proves that our proposed method acts as a fundamental framework of finite-rank and neural network kernel representation.

## 4 NUMERICAL EXPERIMENTS

This section presents experimental results on both synthetic and real data and compares them with several state-of-the-arts, including (i) standard Hawkes process with an exponentially decaying kernel function (`Hawkes`) (Hawkes, 1971); (ii) recurrent marked temporal point processes (`RMTPP`) (Du et al., 2016); and (iii) Neural Hawkes process (`NH`) (Mei & Eisner, 2017). See Appendix B for a detailed review of those existing methods. To perform the experiments, we use the following procedure: Given training data, we first estimate the kernel function to obtain $\hat{k}(\cdot, \cdot)$ by solving the maximum likelihood problem (6) using stochastic gradient descent, as described in Section 2.3. Note that according to (2), the conditional intensity function can be treated as a prediction of the chance of having an event at a given time $t$ after observing the past events. Thus, to evaluate the prediction performance on test data, given a test trajectory, we perform *online prediction* by feeding the past events (in the test trajectory) into evaluating the conditional intensity function according to (3), which gives the conditional probability of a future event given the past observations.

**Performance metrics.** We consider two performance metrics: (i) The performance for synthetic data is evaluated by measuring the *out-of-sample mean-average-error (MAE)* for conditional intensity function. For synthetic data, we know the true kernel, which is denoted as $k^*$. Given a sequence of test data, let $\lambda[k^*]$ be the "true" conditional intensity function defined by (3) using the true kernel $k^*$, and let $\lambda[\hat{k}]$ be an estimated conditional intensity function for the same sequence using the estimated kernel $\hat{k}$. Thus, $\lambda[\hat{k}]$ can be viewed as a probabilistic prediction of the events (since it specifies the likelihood of an event happening given the test trajectory's history) when the kernel $\hat{k}$ is estimated separately from training data. The out-of-sample MAE for one test trajectory is defined as $\int_{\mathcal{X}} |\lambda[k^*](x) - \lambda[\hat{k}](x)| dx$. Then we average this over all test trajectories to obtain our performance measure. (ii) For real data, since we do not know the true intensity function, we report the average *out-of-sample predictive log-likelihood*. Given a test trajectory (information contained in counting measure $\mathbb{N}(x)$), the predictive likelihood for a trajectory using an estimated kernel $\hat{k}$ is given as $\int_{\mathcal{X}} \log \lambda[\hat{k}](x) d\mathbb{N}(x) - \int_{\mathcal{X}} \lambda[k](x) dx$. The average out-of-sample predictive log-likelihood is obtained by average over test trajectories. This has been widely adopted as a metric for real-data (Mei & Eisner, 2017; Omi et al., 2019; Zhang et al., 2019; Zhu et al., 2021a;b;d): the higher the log-likelihood, the better the model predicts. See all experiment configurations in Appendix B.

**Synthetic data.** In the experiment, we assume the background rate is a constant ($\mu = 1$) and focus on recovering both stationary and non-stationary kernel structures. We generate four one-dimensional and three two-dimensional synthetic data sets, which are simulated by a vanilla Hawkes process and our model described in Section 2.2 with randomly initialized parameters. The simulated data is generated by the thinning algorithm described in Algorithm 3 (Appendix A), which is an accept-reject method for simulating a point process (Daley & Vere-Jones, 2008; Gabriel et al., 2013). For ease of comparison, we normalize the time and mark spaces for all data sets to the range from 0 to 100. Each data set is composed of 1,000 event sequences with an averaged length of 121. We fit the models using 80% of the synthetic data set and the remaining 20% as the testing set.

**Kernel recovery.** One of the most important ability of our approach is to recover the true kernel. We report the kernel recovery results for one- and two-dimensional synthetic data sets, shown in Figure 3 and Figure 4, respectively. In Figure 3, we observe that our proposed model can accurately recover the true kernel for one-dimensional data. In contrast, the vanilla Hawkes process with a stationary kernel failed to capture such non-stationarity. We also present results of two-dimensional data in Figure 4, where a continuous one-dimensional mark is introduced to the events. The first three columns demonstrate two-dimensional "slices" of the four-dimensional kernel evaluation, indicating our model is also capable of recovering complex high-dimensional kernel structure. We note that event points are sparsely scattered in the two-dimensional marked-temporal space, where the events' correlation is measured in a high-dimensional (in our example, four-dimensional) kernel space.

**Intensity prediction.** We evaluate the predictive accuracy for conditional intensity $\lambda[\hat{k}](x)$ (3) using estimated kernel $\hat{k}$ from the training data. Note that $\lambda[\hat{k}](x)$ is a probabilistic prediction since it can be viewed as the instantaneous probability of an event given the historical events. The last panel in Figure 3 shows the predicted one-dimensional conditional intensity functions given a sequence randomly selected from the testing set. We compare our predicted $\lambda[\hat{k}](x)$ with the one estimated by a vanilla Hawkes model with stationary kernel. The result shows that the predicted conditional

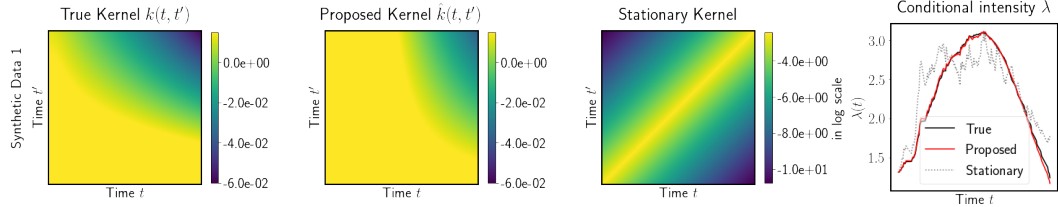

Figure 3: Kernel recovery results on a one-dimensional synthetic data set. The first three panels show the true kernel that generates the data, kernel learned by our model, and kernel learned by a vanilla Hawkes process, respectively. The fourth panel shows the true and predicted conditional intensity functions of a test sequence.

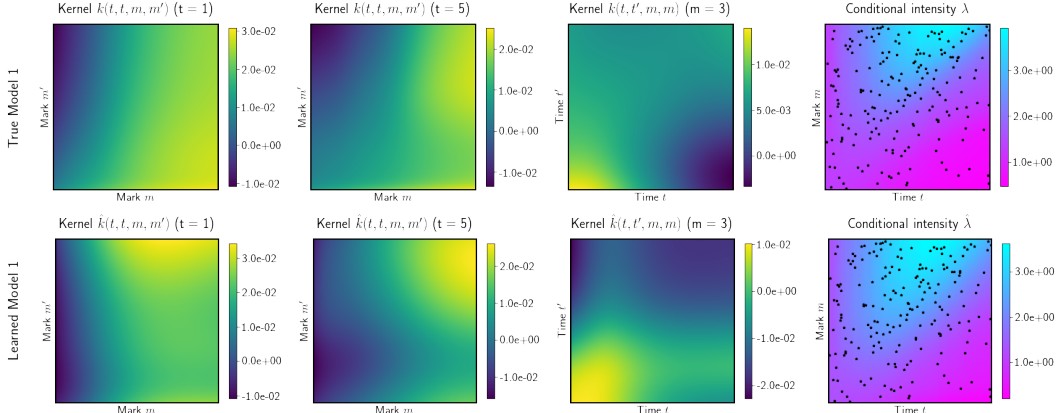

Figure 4: Kernel recovery results on a two-dimensional synthetic data set. Two rows presents the results of the true model and our learned model, respectively. The first three columns show different snapshots of kernel evaluation for each model; the last column shows their corresponding conditional intensity over marked-temporal space given a test sequence, where the black dots indicate the location of the events.

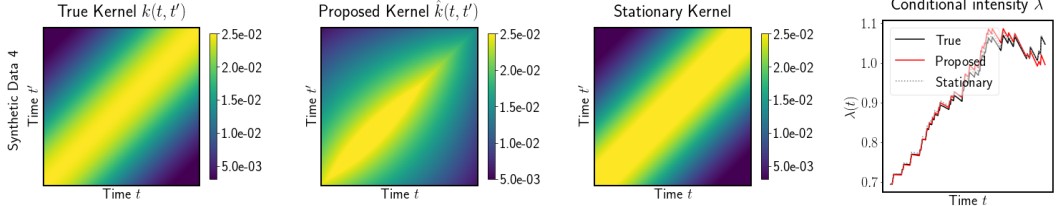

Figure 5: Kernel recovery results on a one-dimensional synthetic data set generated by a vanilla Hawkes process with a *stationary exponentially decaying* kernel. This experiment acts as a sanity check.

intensity $\lambda[\hat{k}](x)$ suggested by our model well matches that using the true kernel $\lambda[k^*](x)$. In contrast, the Hawkes model only provides limited flexibility for modeling the non-stationary process. Two panels in the last column of Figure 4 give another comparison between the true and predicted two-dimensional conditional intensities over marked-temporal space. This result confirms that our model can accurately predict the two-dimensional conditional intensity function. To validate the robustness of our proposed method, we also test our model using another one-dimensional data set generated by a vanilla Hawkes process with a stationary parametric kernel, as shown in Figure 5. Though our model is evidently overparametrized for this data, our method is still able to recover the kernel structure, as well as predict the conditional intensity accurately, which demonstrate the adaptiveness of our model and confirm that our model can approximate the stationary kernel without incurring overfitting. More results of kernel recovery experiments can be found in Appendix C.

Additionally, we compare our method with three other baselines on the synthetic data sets. Figure 6 shows the conditional intensities based on the true kernel $\lambda[k^*](x)$ (solid black lines) and predicted intensity using each method. We observe that our method obtained much better predictive performances compared to other baseline approaches. Table 1 summarizes two performance metrics for these methods: the predictive log-likelihood and the MAE. The result shows that our method greatly outperforms other baseline approaches in both metrics. In particular, the MAE of our method is at least 90% lower than the other methods for both one-dimensional and two-dimensional data sets.

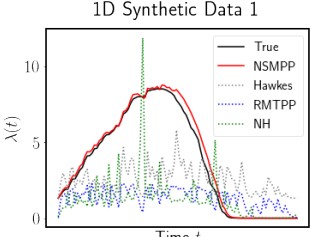 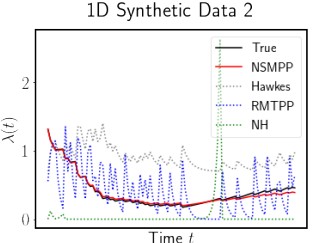 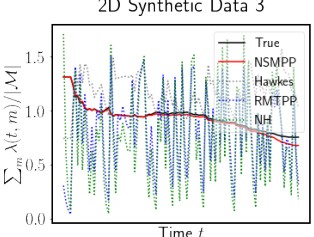

Figure 6: Predicted conditional intensity using our method and other baselines for a sequence selected from test data. We aggregate the conditional intensity in mark space for ease of presentation and visualize the average conditional intensity over time for two-dimensional synthetic data on the third panel.

Table 1: Performance comparison of our method and other existing methods: `Hawkes`, `RMTPP`, and `NH`.

| $\ell$ / MAE[1] | 1D Synthetic 1 | 1D Synthetic 2 | 1D Synthetic 3 | 2D Synthetic 1 | 2D Synthetic 2 | 2D Synthetic 3 | Earthquake (2D) | Robbery (1D) | #Parameters | Training/Testing time[2] |
|---|---|---|---|---|---|---|---|---|---|---|
| NSMPP | **-17.68 / 0.24** | **-14.17 / 0.02** | **-21.94 / 0.02** | **-65.92 / 10.16** | **-87.02 / 0.20** | **-91.12 / 0.31** | **-56.50** / NA | **-74.47** / NA | 171,555 | 0.766 / 0.84 |
| RMTPP | -29.84 / 3.27 | -36.10 / 0.33 | -89.56 / 0.34 | -236.32 / 98.32 | -320.01 / 20.97 | -456.92 / 29.18 | -218.39 / NA | -132.55 / NA | 274,168 | 0.245 / 7.29 |
| NH | -48.34 / 3.42 | -52.97 / 0.41 | -60.10 / 0.44 | -289.10 / 49.21 | -219.74 / 12.45 | -420.00 / 28.99 | -189.39 / NA | -96.10 / NA | 282,755 | 0.204 / 6.09 |
| Hawkes | -24.12 / 2.65 | -77.36 / 0.52 | -61.46 / 0.25 | NA / NA | NA / NA | NA / NA | NA / NA | -197.84 / NA | 2 | 0.021 / <0.01 |

[1] Each table's entry of the data set includes the average predictive log-likelihood ($\ell$) and the MAE.
[2] Training time represents the time for training one batch. Testing (inference) time refers to the time for predicting the conditional intensity function for a test sequence. Time is measured in seconds.

**Real-data results.** Finally, we test our method on two large-scale real data sets: (1) *Atlanta 911 calls-for-service data.* The data set contains the 911 calls-for-service data in Atlanta from 2015 to 2017. We extract 7,831 reported robberies from the data set since robbers usually follow a particular *modus operandi* (M.O.). Each robbery report is associated with a timestamp indicating when the robbery occurred. We consider each series of robberies as a sequence. (2) *Northern California seismic data.* The Northern California Earthquake Data Center (NCEDC) provides public time series data Northern California Earthquake Data Center. UC Berkeley Seismological Laboratory. Dataset (2014) that comes from broadband, short period, strong motion seismic sensors, GPS, and other geophysical sensors. We extract 16,401 seismic records with a magnitude larger than 3.0 from 1978 to 2018 in Northern California and partition the data into multiple sequences every quarter. We fit the models using 80% of the data set and the remaining 20% as the testing set.

In Table 1, we report the *out-of-sample predictive log-likelihood* of each method since the ground truth is not available for real data. We can see that our model attains the highest predictive log-likelihood for both synthetic and real data sets. In particular, the better performance on real data and significantly higher out-of-sample predictive log-likelihood achieved by our approach than other existing methods show that the non-stationary kernel seemingly captures the nature of the real data better in these cases. This shows the merit of our approach for real data where we do not know the ground truth.

**Training / testing time.** In Table 1, we compare the running time of our model and other baselines on one-dimensional synthetic data set. The size and performance of each model are presented in Table 1. The running time refers to wall clock time. We can observe that the training time of our model is similar with other neural point process models (RMTPP and NH). Hawkes process model has the minimum training time because it has only two parameters to be estimated. The testing (inference) time of our model is significantly lower than the other two recurrent neural point process models.

## 5  DISCUSSIONS

We have presented a new non-stationary marked point process model with a general kernel represented by neural network feature maps, motivated by spectral decomposition of the kernel. The flexible kernel with expressive power enabled by neural networks can capture complex dependence across temporal, spatial, and mark spaces, which can be valuable for real-data modeling in various applications. The benefits of our approach are demonstrated by superior out-of-sample predictive log-likelihood on real data. The model can be learned efficiently with stochastic gradient descent. We also develop a theoretical guarantee for maximum likelihood identifiability. Generally, model architectures may well depend on specific tasks. For example, neural networks can be based on CNN if the high dimensional markers are in image-like shapes and can also be LSTM or even BERT if the markers are text-based. Thus the choice of the deep model architecture and optimization algorithm can be more systematically explored, especially the comparison of non-stationary neural kernels to stationary ones. Understanding and characterizing what kernels can be learned through such an approach is left for future study. Also, when extending to high-dimensional mark space, more efficient algorithm is needed for the computation of the log-likelihood.

ACKNOWLEDGEMENT

The work is supported by National Science Foundation (NSF) DMS-2134037. The works of S.Z., H.W., and Y.X. are supported by NSF CAREER Award, CCF CCF-1650913, DMS-1830210, DMS-1938106. The work of X.C. is partially supported by the Alfred P. Sloan Foundation.

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

## A  ALGORITHMS

---

**Algorithm 1:** Stochastic gradient-based learning algorithm

---

**Input:** $X = \{\boldsymbol{x}_j\}_{j=1,\ldots,N}$ is the training set with $N$ sequences; $\eta$ is the number of learning iterations; $\gamma$ is the learning rate; $M$ is the size of a mini-batch;
**Initialization:** model parameters $\boldsymbol{\theta}_0$;
$l \leftarrow 0$;
**while** $l < \eta$ **do**
    Randomly draw $M$ sequences from $X$ denoted as $\widehat{X}_l = \{\boldsymbol{x}_j\}_{j=1,\ldots,M} \subset X$;
    $\boldsymbol{\theta}_{l+1} \leftarrow \boldsymbol{\theta}_l + \gamma \partial\ell/\partial\boldsymbol{\theta}_l$ given $\widehat{X}_l$;
    $l \leftarrow l + 1$;
**end**

---

**Algorithm 2:** Monte Carlo estimation for the integral of conditional intensity in (6)

---

**Input:** $\lambda$ denotes the model that we need to evaluate; $\boldsymbol{x}_j = \{(t_i, m_i)\}_{i=1}^{N_j}$ is an event sequence, where $N_j$ is the number of events in the sequence; $\widetilde{N}$ is the number of samples uniformly drawn from $\mathcal{X}$; $\Lambda$ is the integral of the conditional intensity over the data space;
$\Lambda \leftarrow 0$;
**for** $n = 1, \ldots, \widetilde{N}$ **do**
    Draw $x_n := (t_n, m_n) \sim \mathcal{X}$;
    $\mathcal{H}_{t(x_n)} \leftarrow \{(t_i, m_i) \in \boldsymbol{x}_j : t_i < t_n\}$;
    $\Lambda \leftarrow \Lambda + \lambda_j(x_n)$ given $\mathcal{H}_{t(x_n)}$;
**end**
$\Lambda \leftarrow |\mathcal{X}|\Lambda/\widetilde{N}$;

---

**Algorithm 3:** Efficient thinning algorithm for simulating point process

---

**input** $\boldsymbol{\theta}, T, \mathcal{M}$;
**output** A set of events $\mathcal{H}_t$ ordered by time.;
Initialize $\mathcal{H}_t = \emptyset, t = 0, m \sim \texttt{uniform}(\mathcal{M})$;
**while** $t < T$ **do**
    Sample $u \sim \texttt{uniform}(0, 1)$; $m \sim \texttt{uniform}(\mathcal{M})$; $D \sim \texttt{uniform}(0, 1)$;
    $x' \leftarrow (t, m')$; $\bar{\lambda} \leftarrow \lambda(x'|\mathcal{H}_t)$;
    $t \leftarrow t - \ln u/\bar{\lambda}$;
    $x \leftarrow (t, m)$; $\widetilde{\lambda} \leftarrow \lambda(x|\mathcal{H}_t)$;
    **if** $D\bar{\lambda} > \widetilde{\lambda}$ **then**
        $\mathcal{H}_t \leftarrow \mathcal{H}_t \cup \{(t, m)\}$; $m' \leftarrow m$;
    **end**
**end**

---

## B  EXPERIMENTAL SETTING AND BASELINE METHODS

Now we describe the experiment configurations: We set the rank $R = 5$ in our setting. We consider the shared network that summarizes input data into a hidden embedding to be a fully connected three-layer network and the sub-network to be a fully connected network with two hidden layers. The width of the hidden layers in the shared network is $n = 128$, and the width of the input layers in sub-networks (or the output layer in the shared network) is $p = 10$. We adopt the SoftPlus $f(x) = 1/\log(1 + \exp(x))$ as the activation function of each layer in the network. To learn the model's parameters, we adopt the Adam optimizer Kingma & Ba (2014) with a constant learning rate of $10^{-2}$ and the batch size is 32. All experiments are performed on Google Colaboratory (Pro

version) with 12GB RAM and dual-core Intel processors, which speed up to 2.3 GHz (without GPU). Codes to reproduce the experimental results are publicly available[1].

This study considers three baseline methods: (1) Standard Hawkes process with an exponentially decaying kernel function (`Hawkes`): it specifies the conditional intensity function as $\lambda(t) = \mu + \alpha \sum_{t_j < t} \beta \exp\{-\beta(t - t_j)\}$, where parameters $\mu, \alpha, \beta$ can be estimated via maximizing likelihood (Hawkes, 1971); (2) Recurrent marked temporal point processes (`RMTPP`): it assumes the conditional intensity function $\lambda(t) = \exp\left(\boldsymbol{v}^\top \boldsymbol{h}_j + \omega(t - t_j) + b\right)$, where the $j$-th hidden state $\boldsymbol{h}_j$ in the RNN represents the history influence up to the nearest happened event $j$, and $w(t - t_j)$ represents the current influence; the $\boldsymbol{v}, \omega, b$ are trainable parameters (Du et al., 2016); and (3) Neural Hawkes process (`NH`): it specifies the conditional intensity function as $\lambda^*(t) = f(\boldsymbol{\nu}^\top \boldsymbol{h}_t)$, where $\boldsymbol{h}_t$ is the hidden state of a continuous-time LSTM up to time $t$ representing the history influence, and the $f(\cdot)$ is a SoftPlus function which ensures the positive output given any input (Mei & Eisner, 2017).

We note that, unlike the standard Hawkes process and our `NSMPP`, `RMTPP` and `NH` do not parameterize the kernel function directly; instead, they aim to model the conditional intensity using an LSTM-based structure. Particularly, these models pass the history information sequentially via a hidden state, where the recent memory will override the long-term memory. This has led `RMTPP` and `NH` to "overemphasize" the recent events and therefore assume the temporal correlation would monotonically decrease over time. In addition, `RMTPP` and `NH` can only deal with one-dimensional categorical marks, while our model can be extended to high-dimensional continuous mark space. To ensure comparability, we only consider one- and two-dimensional event consisting of time and mark in our experiment ($d \in \{1, 2\}$). For `RMTPP` and `NH`, the event's mark will be discretized and treated as categorical input.

## C    ADDITIONAL EXPERIMENTAL RESULTS

In this section, we present additional experiments to demonstrate the robustness of our approach by considering fitting to stationary exponentially decaying kernel and other non-stationary kernels and by varying the neural network architecture and training sample sizes.

Figure 7 and Figure 8 show the results on one- and two-dimensional data sets with *non-stationary kernels* generated by model described in Section 2.2. Note that compared to the Hawkes models with a stationary kernel, our model can capture the non-stationarity of the kernel and predict the conditional intensity function $\lambda$ more accurately. Besides, our proposed method also recovers the kernel structure and predicts $\lambda$ well in high-dimensional space. Note that the event points are more densely scattered in the areas with high intensity than those with much lower ones.

Figure 9 and Figure 10 study the effect of increased model size and decreased training sample sizes to the learning of our model. The results of these two ablation studies show that our proposed model works consistently well without overfitting after increasing the model size or decreasing the training sample size in both the stationary and non-stationary cases.

## D    MLE GUARANTEE UNDER FINITE DIMENSIONAL FUNCTIONAL REPRESENTATION

We notice that in practice, the kernel function is usually restricted to some function family space for the problem to be well-defined (given a finite number of events ). In this section, we study such a set-up, where the target feature function belongs to a certain parametric function class with a finite number of parameters. This includes, for instance, spline functions, functions represented by Fourier basis, and by neural networks – the main interest of this paper. We are particularly interested in neural networks due to its strong expressive power of functional representation.

We start from a general framework of feature function basis representation. Assume that the feature functions of the kernel can be well-approximated by a linear combination of basis functions: $b_i(x)$ :

---

[1] https://github.com/meowoodie/Neural-Spectral-Marked-Point-Processes

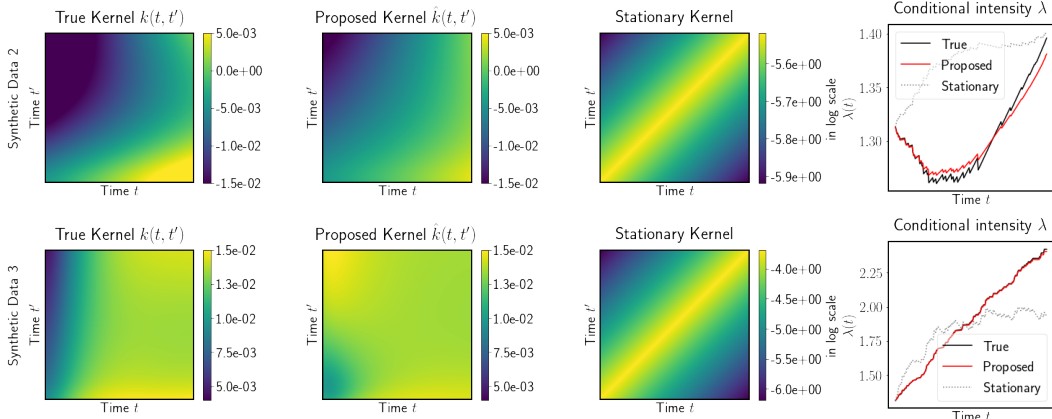

Figure 7: Additional kernel recovery results for two other one-dimensional synthetic data sets. The first three columns show the true kernel that generates the data, kernel learned by our model, and kernel learned by a Hawkes process, respectively. The fourth column shows the true and predicted conditional intensity functions for a test sequence.

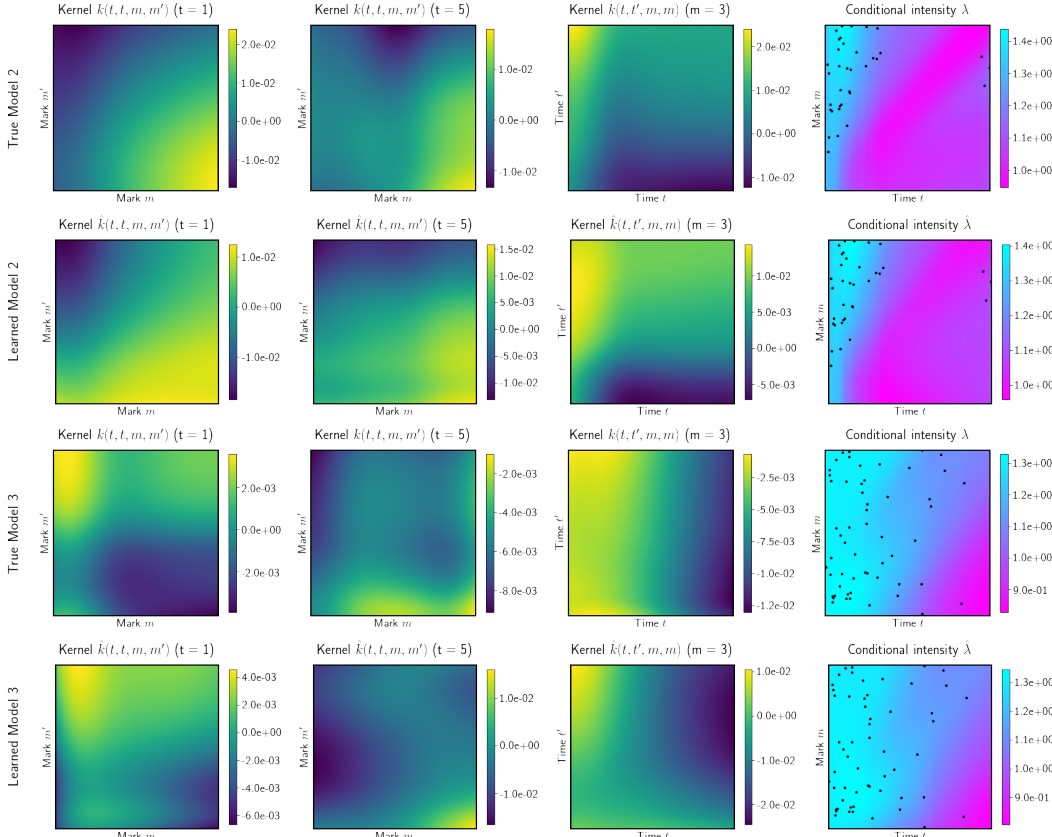

Figure 8: Additional kernel recovery results for two other two-dimensional synthetic data sets. The first and third rows show the true models and the second and fourth rows show the learned models. The first three columns show different snapshots of kernel evaluation for each model; the last column shows their corresponding conditional intensity over marked-temporal space given a test sequence, where the black dots indicate the location of the events.

$\mathcal{X} \to \mathbb{R}, i = 1, \ldots, S$:

$$\psi_r(x) = \sum_{i=1}^{S} \alpha_{ri} b_i(x), \quad \phi_r(x) = \sum_{i=1}^{S} \beta_{ri} b_i(x), \quad r = 1, \cdots, R.$$

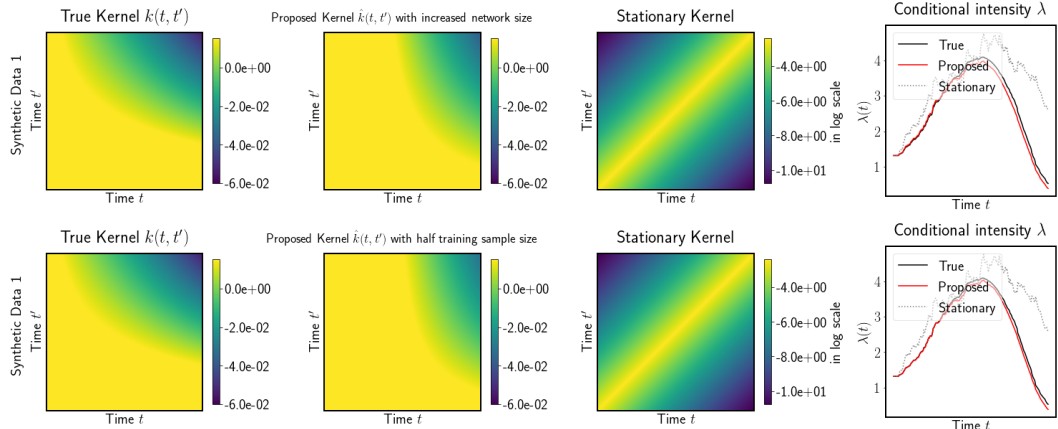

Figure 9: Ablation studies on one-dimensional synthetic data sets used in Figure 3. "Proposed kernel with increased network size" refers to the model with one more hidden layer and doubled layer width in the sub-networks ($p = 20$); "Proposed kernel with half training sample size" refers to the model with default architecture but trained with only a half of the training samples.

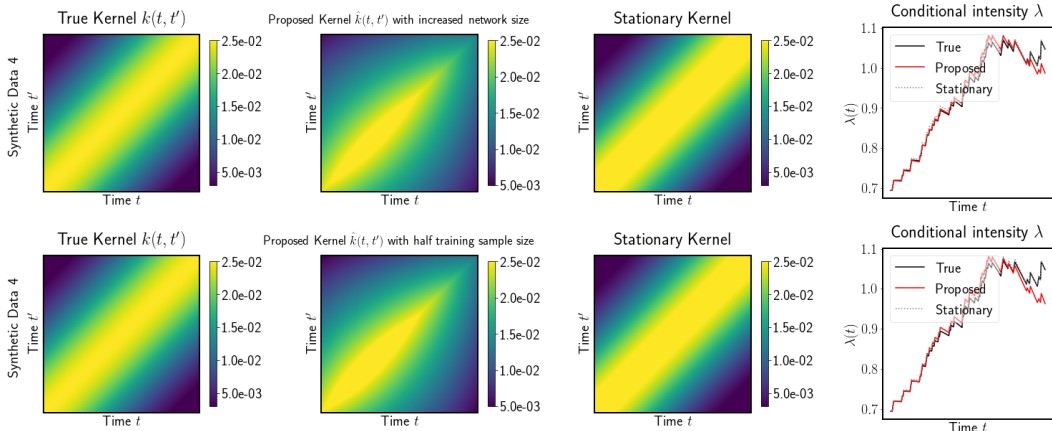

Figure 10: Ablation studies on one-dimensional synthetic data set used in Figure 5.

Then the kernel function in (5) can be written as $k_A(x', x) = b(x')^T A b(x)$, where $b(x) = (b_1(x), \cdots, b_S(x))^T$, and the $(p, q)$-th entry of the matrix $A$ is given by

$$A_{pq} = \sum_{r=1}^{R} \nu_r \alpha_{rp} \beta_{rq}.$$

Here we assume each matrix $A$ corresponds to a unique model, i.e. for any $A' \neq A$, there exists $x', x \in \mathcal{X}$ with $t(x') < t(x)$, $k_A(x', x) \neq k_{A'}(x', x)$. Under such a parametrization, we can write the intensity function in (4) as

$$\lambda_A(x) := \lambda[k_A](x) = \mu + \int_{\mathcal{X}_{t(x)}} b(x')^T A b(x) d\mathbb{N}(x') = \mu + \langle \eta(x), A \rangle, \tag{10}$$

which is linear in the vectorized $A$. Here $\langle \cdot, \cdot \rangle$ is the Frobenius inner product of matrices, and $\eta(x) \in \mathbb{R}^{S \times S}$ is conditioned on $\mathcal{H}_{t(x)}$ with the $(p, q)$-th entry

$$\eta_{pq}(x) = \int_{\mathcal{X}_{t(x)}} b_p(x') b_q(x) d\mathbb{N}(x').$$

Now the influence kernel estimation problem has been reduced to the problem of estimating the $S$-by-$S$ matrix $A$. We will estimate the coefficient matrix $A$ by the Maximum Likelihood Estimator

(MLE) in the set of low-rank matrices $\mathcal{A} = \{A \in \mathbb{R}^{S \times S} : k_A \in \mathcal{K}\}$ (recall that $\mathcal{K}$ is the family of kernels which admits the finite-rank decomposition as in (5). By saying $\mathcal{A}$ consists of low-rank matrices, we implicitly assume that $R \ll S$). Also we denote the family of kernel functions that can be represented by the chosen basis functions as

$$\mathcal{K}_{\text{finite}} = \{k_A : A \in \mathcal{A}\}.$$

The MLE is defined as

$$\widehat{A}_{\text{MLE}} = \arg\max_{A \in \mathcal{A}} \ell_A,$$

where the log-likelihood $\ell_A$ is the corresponding variant of (6),

$$\ell_A = \frac{1}{M} \left( \sum_{j=1}^{M} \int_{\mathcal{X}} \log\left(\mu + \langle \eta_j(x), A \rangle\right) d\mathbb{N}_j(x) - \sum_{j=1}^{M} \int_{\mathcal{X}} \left(\mu + \langle \eta_j(x), A \rangle\right) dx \right).$$

- With orthonormal bases, the classical theory has that the recovery of $A$ will ensure the recovery of the original kernel function. With a set of over-complete bases, $R$ is less than $S$ (the number of bases function), and then $A$ is a low-rank matrix, for which case we provide a theory for the recovery of $A$ via MLE (Theorem D.2).
- More generally, the above framework contains other constructions of $b_i(x)$'s. For example, if $b_i(x)$ are random features, then $\alpha_{ri}$ and $\beta_{ri}$ can be viewed as weights to combine features. Random feature model can be naturally viewed as a neural network with one hidden layer, that is, $b_i(x) = \sigma(w_i^T x)$ is the activation on the $i$-th hidden nodes, and $\sum_i \alpha_i b_i(x)$ gives the second layer output function. Here, $w_i$ are weights in the first layer, and linear combination coefficients $\alpha_i$ are weights in the second layer. Thus, the random feature model corresponds to only training the second layer weights, leaving the 1st layer as randomly initialized. In this case, $S$ is the number of hidden neurons, and $R$ can be interpreted as the number of heads in an attention model.

**A few consequences.** We derive a few necessary basic results based on the model parametrization for presenting the results. Recall that $\eta(x) \in \mathbb{R}^{S \times S}$ depends only on data and basis but not the coefficient matrix $A$.

- For each trajectory $j$, the intensity under parameter $A$ at $x$ has partial derivative

$$\frac{\partial \lambda_{A,j}(x)}{\partial A_{pq}} = \eta_{pq,j}(x).$$

- The score function, i.e., the partial gradient of the log-likelihood function with respect to the coefficient matrix $A$, is given by

$$\frac{\partial \ell_A}{\partial A_{pq}} = \frac{1}{M} \sum_{j=1}^{M} \int_{\mathcal{X}} \lambda_{A,j}(x)^{-1} \eta_{pq,j}(x)(d\mathbb{N}_j(x) - \lambda_{A,j}(x)dx).$$

- The Hessian matrix of the log-likelihood function is given by

$$\frac{\partial^2 \ell_A}{\partial A_{pq} \partial A_{rs}} = -\frac{1}{M} \sum_{j=1}^{M} \int_{\mathcal{X}} \lambda_{A,j}^{-2}(x) \eta_{pq,j}(x) \eta_{rs,j}(x) d\mathbb{N}_j(x).$$

Next, we provide some analysis for the MLE with possible model misspecification, as the true kernel $k^*$ may not fall into $\mathcal{K}_{\text{finite}}$.

**Theorem D.1** (Distance between the true kernel and the optimal fit). *Let $\widetilde{A} \in \mathcal{A}$ be the one which maximizes the expected log-likelihood function, i.e.*

$$\widetilde{A} = \arg\max_{A \in \mathcal{A}} \mathbb{E}\left(\ell_A\right). \tag{11}$$

*Under Assumption 3.1, let the $\ell_2$-norm of a kernel be*

$$\|k\|_2^2 = \int_{\mathcal{X}} \int_{\mathcal{X}_{t(x)}} k(x', x)^2 dx' dx. \tag{12}$$

*Then we have*

$$\|k^* - k_{\widetilde{A}}\|_2^2 \le \frac{c_2^5 |\mathcal{M}|T + c_2^4}{c_1^4} \exp(2(c_2 - c_1)|\mathcal{M}|T) D(k^*, \mathcal{K}_{\text{finite}})^2,$$

*where $D(k^*, \mathcal{K}_{\text{finite}})$ is the $\ell_2$-distance between the true kernel and the set $\mathcal{K}_{\text{finite}}$,*

$$D(k^*, \mathcal{K}_{\text{finite}}) = \min_{k \in \mathcal{K}_{\text{finite}}} \|k^* - k\|_2.$$

*Remark* D.1. This theorem holds true for any $k^* \in \overline{\mathcal{K}}$ without the rank-$R$ assumption on kernels. In this case, both feature function approximation by basis and low-rank approximation of the kernel contribute to the estimation error associated with model misspecification.

Next, similar to the classic asymptotic normality of the MLE under model misspecification (White, 1982), we have the following result for the low-rank MLE.

**Theorem D.2** (Asymptotic normality of low-rank MLE). *Under Assumption 3.1, Assumption E.1, let the singular value decomposition of $\widetilde{A}$ be $\widetilde{A} = U\Lambda V^T$. Let $\widetilde{I}$ be the expected Hessian matrix of the log-likelihood at $\widetilde{A}$,*

$$\widetilde{I} = \mathbb{E}\left(\frac{\partial^2 \ell_A}{\partial \mathrm{vec}(A)\partial \mathrm{vec}(A)^T}\right)\bigg|_{A=\widetilde{A}} = -\mathbb{E}\left(\int_{\mathcal{X}} \lambda_{\widetilde{A}}^{-2}\mathrm{vec}(\eta(x))\mathrm{vec}(\eta(x))^T d\mathbb{N}(x)\right),$$

*where the matrices are vectorized by concatenating their columns, and let $\widetilde{J}$ be the covariance matrix of a single trajectory's score function at $\widetilde{A}$,*

$$\widetilde{J} = \mathrm{Cov}\left(\frac{\partial \ell_A}{\partial \mathrm{vec}(A)}, \frac{\partial \ell_A}{\partial \mathrm{vec}(A)^T}\right)\bigg|_{A=\widetilde{A}},$$

$\widetilde{G} \in \mathbb{R}^{S\times S}$ *be the expected score at $\widetilde{A}$,*

$$\widetilde{G} = \mathbb{E}\left(\frac{\ell_A}{\partial A}\right)\bigg|_{A=\widetilde{A}}.$$

*Let $F = (\mathbb{I}_S \otimes U, V \otimes \mathbb{I}_S) \in \mathbb{R}^{S^2 \times 2SR}$ where $\otimes$ is the Kronecker product, $\mathbb{I}_S$ is the identity matrix of size $S$,*

$$\widetilde{C} = (\widetilde{A}^\dagger \otimes \widetilde{G})Q_{S,S} + ((\widetilde{A}^\dagger \otimes \widetilde{G})Q_{S,S})^T,$$

*where $\dagger$ represents pseudo-inverse and $Q_{a,b} \in \mathbb{R}^{ab\times ab}$ is the permutation matrix such that $\mathrm{vec}(P^T) = Q_{a,b}\mathrm{vec}(P)$ for any $a$-by-$b$ matrix $P$. If $F^T(\widetilde{I} + \widetilde{C})F$ shares the same null-space with $F$, then the low-rank estimator $\widehat{A}_{\mathrm{MLE}}$, solved from the constrained maximum likelihood problem, satisfies*

$$\sqrt{M}(\mathrm{vec}(\widehat{A}_{\mathrm{MLE}}) - \mathrm{vec}(\widetilde{A})) \to \mathcal{N}(0, F(F^T(\widetilde{I} + \widetilde{C})F)^\dagger F^T \widetilde{J}F(F^T(\widetilde{I} + \widetilde{C})F)^\dagger F),$$

*when $M \to \infty$.*

*Remark* D.2 (Parameter recovery guarantee). The result shows that the maximum likelihood estimate for the kernel matrix $A$ converges to the optimal fit $\widetilde{A}$, and the matrices $\widetilde{I}, \widetilde{J}, \widetilde{C}$ captures the residual variance; $F$ projects the variance caused by $\widetilde{I}, \widetilde{J}, \widetilde{C}$ onto the tangent space of the low-rank manifold at $\widetilde{A}$. In practice, the variance term can be estimated empirically; $\widetilde{J}$ can be estimated from the empirical covariance matrix of the score function of each trajectory, while the other matrices can be approximated by the properties of the log-likelihood function at $\widehat{A}_{\mathrm{MLE}}$. If the true kernel falls into the kernel family $\mathcal{K}_{\mathrm{finite}}$, both $\widetilde{I}, \widetilde{J}$ will equal to the Fisher Information, $\widetilde{G}, \widetilde{C}$ will vanish, which greatly simplify the process of estimating the variance. But with the presence of model misspecification, this is generally not the case.

# E    PROOFS

## E.1    PROOF OF LEMMA 3.1

*Proof of Lemma 3.1.* For the $j$-th trajectory, define $\ell_j[k]$ as

$$\ell_j[k] = \int_{\mathcal{X}} \log \lambda_j[k](x)d\mathbb{N}_j(x) - \int_{\mathcal{X}} \lambda_j[k](x)dx, \tag{13}$$

then $\ell[k] = \frac{1}{M}\sum_j \ell_j[k]$. Let $\lambda_j^* = \lambda_j[k^*]$, $\tilde{\lambda}_j = \lambda_j[\tilde{k}]$, where the conditional intensity $\lambda[k]$ is defined as in (4), $k^*$ and $\tilde{k}$ are the true kernel and the perturbed one respectively. Then we have

$$\ell_j[\tilde{k}] - \ell_j[k^*] = \int_{\mathcal{X}} (\log \tilde{\lambda}_j(x) - \log \lambda_j^*(x))d\mathbb{N}_j(x) - \int_{\mathcal{X}} (\tilde{\lambda}_j(x) - \lambda_j^*(x))dx \tag{14}$$

By (4), $\lambda_j[k]$ is linear with respect to perturbation in $k$, that is,

$$\lambda_j[k](x) = \nu + \int_{\mathcal{X}_{t(x)}} k(x', x) d\mathbb{N}_j(x').$$

We then have that

$$\tilde{\lambda}_j(x) - \lambda_j^*(x) = \int_{\mathcal{X}_{t(x)}} (\tilde{k}(x', x) - k^*(x', x)) d\mathbb{N}_j(x') = \delta\lambda_j(x), \qquad (15)$$

where the last equality is by definition of $\delta\lambda_j$.

Back to (14), the second term

$$\int_{\mathcal{X}} (\tilde{\lambda}_j(x) - \lambda_j^*(x)) dx = \int_{\mathcal{X}} \delta\lambda_j(x) dx; \qquad (16)$$

The first term under Taylor expansion of $\log$ has that, for each $x \in \mathcal{X}$,

$$\log \tilde{\lambda}_j(x) - \log \lambda_j^*(x) = \frac{1}{\lambda_j^*(x)} \delta\lambda_j(x) - \frac{1}{2\xi_j(x)^2} (\delta\lambda_j(x))^2,$$

where $\xi_j(x)$ takes value between $\tilde{\lambda}_j(x)$ and $\lambda_j^*(x)$. By Assumption 3.1, both $\tilde{\lambda}_j(x)$ and $\lambda_j^*(x)$ are strictly positive and upper-bounded by $c_2$, and thus $0 < \xi_j(x) \le c_2$. This means that

$$\frac{1}{2\xi_j(x)^2} (\delta\lambda_j(x))^2 \ge \frac{1}{c_2^2} (\delta\lambda_j(x))^2, \quad \forall x \in \mathcal{X}.$$

Thus, the 1st term in (14) satisfies

$$\int_{\mathcal{X}} (\log \tilde{\lambda}_j(x) - \log \lambda_j^*(x)) d\mathbb{N}_j(x) = \int_{\mathcal{X}} \delta\lambda_j(x) \frac{d\mathbb{N}_j(x)}{\lambda_j^*(x)} - \int_{\mathcal{X}} \frac{1}{2\xi_j(x)^2} (\delta\lambda_j(x))^2 d\mathbb{N}_j(x)$$

$$\le \int_{\mathcal{X}} \delta\lambda_j(x) \frac{d\mathbb{N}_j(x)}{\lambda_j^*(x)} - \frac{1}{2c_2^2} \int_{\mathcal{X}} (\delta\lambda_j(x))^2 d\mathbb{N}_j(x). \qquad (17)$$

Putting together (16) and (17),

$$\ell_j[\tilde{k}] - \ell_j[k^*] \le \int_{\mathcal{X}} \delta\lambda_j(x) \frac{d\mathbb{N}_j(x)}{\lambda_j^*(x)} - \frac{1}{2c_2^2} \int_{\mathcal{X}} (\delta\lambda_j(x))^2 d\mathbb{N}_j(x) - \int_{\mathcal{X}} \delta\lambda_j(x) dx$$

$$= \int_{\mathcal{X}} \delta\lambda_j(x) \left( \frac{d\mathbb{N}_j(x)}{\lambda_j^*(x)} - dx \right) - \frac{1}{2c_2^2} \int_{\mathcal{X}} (\delta\lambda_j(x))^2 d\mathbb{N}_j(x).$$

The above holds for each $j$, and taking the average over the $M$ trajectories proves the lemma. $\square$

### E.2  PROOF OF THEOREM 3.2

**Lemma E.1** (lower bound for KL-divergence). *Under Assumption 3.1,*

$$\mathbb{E}\left( \ell[k^*] - \ell[\tilde{k}] \right) \ge \frac{c_1^2}{2c_2^2} \exp\left( -(c_2 - c_1)|\mathcal{M}|T \right) \|\delta k\|_2^2,$$

*holds true for any $\tilde{k} \in \overline{\mathcal{K}}$ and $\delta k = \tilde{k} - k^*$ (the $\ell_2$-norm is defined as (12).)*

*Proof.* By Lemma 3.1, since all trajectories are i.i.d., and for each trajectory the conditional expectation

$$\mathbb{E}\left( \frac{d\mathbb{N}(x)}{\lambda[k^*](x)} \middle| \mathcal{H}_{t(x)} \right) = dx,$$

there is

$$\mathbb{E}\left( \ell[k^*] - \ell[\tilde{k}] \right) \ge \mathbb{E}\left( \int_{\mathcal{X}} \delta\lambda(x) \left( \frac{d\mathbb{N}(x)}{\lambda[k^*](x)} - dx \right) + \frac{1}{2c_2^2} \int_{\mathcal{X}} (\delta\lambda(x))^2 d\mathbb{N}(x) \right)$$

$$= \frac{1}{2c_2^2} \mathbb{E}\left( \int_{\mathcal{X}} (\delta\lambda(x))^2 d\mathbb{N}(x) \right).$$

Following Assumption 3.1

$$\frac{1}{2c_2^2}\mathbb{E}\left(\int_{\mathcal{X}}(\delta\lambda(x))^2 d\mathbb{N}(x)\right) = \frac{1}{2c_2^2}\mathbb{E}\left(\int_{\mathcal{X}}(\delta\lambda(x))^2\lambda[k^*](x)dx\right) \geq \frac{c_1}{2c_2^2}\mathbb{E}\left(\int_{\mathcal{X}}(\delta\lambda(x))^2 dx\right).$$

Next we lower bound the term above by taking the integral over the event space $\mathcal{E} = \bigsqcup_{i=0}^{\infty}\mathcal{E}_i$ of trajectories, where for each $i$,

$$\mathcal{E}_i = \{(x_1, x_2, \cdots, x_i) \in \mathcal{X}^i, t(x_1) < \cdots < t(x_i)\} \subset \mathcal{X}^i$$

consists of all the trajectories with exactly $i$ events. For each $\mathcal{H}_T \in \mathcal{E}$, let $\mathbb{N}$ be the associated counting measure, the probability density of $\mathcal{H}_T$

$$\rho(\mathcal{H}_T) = \exp\left(\int_{\mathcal{X}}\log\lambda[k^*]d\mathbb{N}(x) - \int_{\mathcal{X}}\lambda[k^*](x)dx\right) \geq \exp(|\mathcal{H}_T|\log c_1 - |\mathcal{M}|Tc_2) =: \underline{\rho}(\mathcal{H}_T).$$

$\mathbb{E}\left(\int_{\mathcal{X}}(\delta\lambda(x))^2 dx\right)$ can be lower bounded by taking the integral over the lower bound of probability density $\rho$,

$$\frac{c_1}{2c_2^2}\mathbb{E}\left(\int_{\mathcal{X}}(\delta\lambda(x))^2 dx\right) \geq \frac{c_1}{2c_2^2}\sum_{i=0}^{\infty}\int_{\mathcal{E}_i}\left(\int_{\mathcal{X}}(\delta\lambda(x))^2 dx\right)\underline{\rho}(\mathcal{H}_T)dx_1 dx_2\cdots dx_i,$$

and by the equivalence over ordering of $x_1, \cdots, x_i$,

$$\frac{c_1}{2c_2^2}\sum_{i=0}^{\infty}\int_{\mathcal{E}_i}\left(\int_{\mathcal{X}}(\delta\lambda(x))^2 dx\right)\underline{\rho}(\mathcal{H}_T)dx_1 dx_2\cdots dx_i$$

$$= \frac{c_1}{2c_2^2}\sum_{i=0}^{\infty}\frac{1}{i!}\int_{\mathcal{X}^i}\left(\int_{\mathcal{X}}(\delta\lambda(x))^2 dx\right)\underline{\rho}(\mathcal{H}_T)dx_1 dx_2\cdots dx_i. \qquad (*)$$

Then we substitute $\delta\lambda(x)$ with $\int_{\mathcal{X}_{t(x)}}\delta k(x', x)d\mathbb{N}(x')$ and change the order of integrals, $(*)$ equals to

$$= \frac{c_1}{2c_2^2}\sum_{i=0}^{\infty}\frac{1}{i!}\int_{\mathcal{X}^i}\left(\int_{\mathcal{X}}\int_{\mathcal{X}_{t(x)}}\int_{\mathcal{X}_{t(x)}}\delta k(x', x)\delta k(x'', x)d\mathbb{N}(x')d\mathbb{N}(x'')dx\right)\underline{\rho}(\mathcal{H}_T)dx_1 dx_2\cdots dx_i$$

$$= \frac{c_1}{2c_2^2}\sum_{i=0}^{\infty}\frac{1}{i!}\int_{\mathcal{X}}\int_{\mathcal{X}^i}\int_{\mathcal{X}_{t(x)}}\int_{\mathcal{X}_{t(x)}}\delta k(x', x)\delta k(x'', x)\underline{\rho}(\mathcal{H}_T)d\mathbb{N}(x')d\mathbb{N}(x'')dx_1 dx_2\cdots dx_i dx.$$

For $x' \neq x''$, $d\mathbb{N}(x')d\mathbb{N}(x'') = 1$ when two of $x_1, \cdots, x_i$ equal to $x', x''$. For $x' = x''$, $d\mathbb{N}(x')d\mathbb{N}(x'') = 1$ when one of $x_1, \cdots, x_i$ equals to $x' = x''$. Again by the equivalence over ordering of $x_1, \cdots, x_i$, we assume $x' = x_1, x'' = x_2$ when $x' \neq x''$, or $x' = x'' = x_1$, $(*)$ equals to

$$\frac{c_1}{2c_2^2}\sum_{i=2}^{\infty}\frac{i(i-1)}{i!}\int_{\mathcal{X}}\int_{\mathcal{X}^{i-2}}\int_{\mathcal{X}_{t(x)}}\int_{\mathcal{X}_{t(x)}}\delta k(x', x)\delta k(x'', x)\underline{\rho}(\mathcal{H}_T)dx'dx''dx_3\cdots dx_i dx$$

$$+ \frac{c_1}{2c_2^2}\sum_{i=1}^{\infty}\frac{i}{i!}\int_{\mathcal{X}}\int_{\mathcal{X}^{i-1}}\int_{\mathcal{X}_{t(x)}}\delta k(x', x)\delta k(x', x)\underline{\rho}(\mathcal{H}_T)dx'dx_2\cdots dx_i dx$$

For the first row, we take the integral over $x_3, \cdots, x_i$ and for the second row, we take the integral over $x_2, \cdots, x_i$, $(*)$ equals to

$$= \frac{c_1}{2c_2^2}\sum_{i=2}^{\infty}\frac{i(i-1)}{i!}\int_{\mathcal{X}}\int_{\mathcal{X}_{t(x)}}\int_{\mathcal{X}_{t(x)}}\delta k(x', x)\delta k(x'', x)|\mathcal{M}|^{i-2}T^{i-2}c_1^i\exp(-c_2|\mathcal{M}|T)dx'dx''dx$$

$$+ \frac{c_1}{2c_2^2}\sum_{i=1}^{\infty}\frac{i}{i!}\int_{\mathcal{X}}\int_{\mathcal{X}_{t(x)}}\delta k(x', x)\delta k(x', x)|\mathcal{M}|^{i-1}T^{i-1}c_1^i\exp(-c_2|\mathcal{M}|T)dx'dx$$

$$= \frac{c_1}{2c_2^2}\sum_{i=2}^{\infty}\frac{i(i-1)}{i!}\int_{\mathcal{X}}|\mathcal{M}|^{i-2}T^{i-2}c_1^i\exp(-c_2|\mathcal{M}|T)\left(\int_{\mathcal{X}_{t(x)}}\delta k(x', x)dx'\right)^2 dx$$

$$+ \frac{c_1}{2c_2^2} \int_{\mathcal{X}} \int_{\mathcal{X}_{t(x)}} \delta k(x',x) \delta k(x',x) \sum_{i=1}^{\infty} \frac{1}{(i-1)!} |\mathcal{M}|^{i-1} T^{i-1} c_1^i \exp(-c_2|\mathcal{M}|T) dx' dx$$

Note that the first term is non-negative.

$$
\begin{aligned}
(*) &\geq \frac{c_1}{2c_2^2} \int_{\mathcal{X}} \int_{\mathcal{X}_{t(x)}} \delta k(x',x) \delta k(x',x) \sum_{i=1}^{\infty} \frac{1}{(i-1)!} |\mathcal{M}|^{i-1} T^{i-1} c_1^i \exp(-c_2|\mathcal{M}|T) dx' dx \\
&= \frac{c_1^2}{2c_2^2} \int_{\mathcal{X}} \int_{\mathcal{X}_{t(x)}} \delta k(x',x) \delta k(x',x) \exp(-(c_2 - c_1)|\mathcal{M}|T) dx' dx \\
&= \frac{c_1^2}{2c_2^2} \exp(-(c_2 - c_1)|\mathcal{M}|T) \|\delta k\|_2^2.
\end{aligned}
$$

$\square$

*Proof of Theorem 3.2.* This follows immediately from Lemma E.1. $\square$

### E.3 PROOF OF THEOREM D.1

**Lemma E.2.** *Under Assumption 3.1, for any $k \in \overline{\mathcal{K}}$,*

$$\mathbb{E} \left( \ell[k^*] - \ell[k] \right) \leq \frac{c_2^3 |\mathcal{M}|T + c_2^2}{2c_1^2} \exp((c_2 - c_1)|\mathcal{M}|T) \|k - k^*\|_2^2,$$

*Proof.* Similar to the proof of Lemma 3.1 and Theorem 3.2, for a single trajectory and any $k \in \overline{\mathcal{K}}$ let

$$\delta k(x',x) = k(x',x) - k^*(x',x), \forall x', x \in \mathcal{X}, t(x') < t(x),$$

$$\delta \lambda(x) = \int_{\mathcal{X}_{t(x)}} \delta k(x',x) d\mathbb{N}(x').$$

There is

$$
\begin{aligned}
\ell[k^*] - \ell[k] &= \int_{\mathcal{X}} \delta \lambda(x) dx - \int_{\mathcal{X}} \log \left( \frac{\lambda[k](x)}{\lambda[k^*](x)} \right) d\mathbb{N}(x) \\
&= \int_{\mathcal{X}} \delta \lambda(x) dx - \int_{\mathcal{X}} \log \left( 1 + \frac{\delta \lambda(x)}{\lambda[k^*](x)} \right) d\mathbb{N}(x) \\
&= \int_{\mathcal{X}} \delta \lambda(x) \left( dx - \frac{d\mathbb{N}(x)}{\lambda[k^*](x)} \right) + \frac{1}{2} \frac{\delta \lambda(x)^2}{\bar{\lambda}(x)^2} d\mathbb{N}(x),
\end{aligned}
$$

for some $\bar{\lambda}(x)$ determined by $\mathcal{H}_{t(x)}$ such that $\bar{\lambda}(x)$ is in between $\lambda[k^*](x)$ and $\lambda[k](x)$ for all $x \in \mathcal{X}$. Then since the expectation of $dx - d\mathbb{N}(x)/\lambda[k^*](x)$ is 0,

$$
\begin{aligned}
\mathbb{E} \left( \ell[k^*] - \ell[k] \right) &= \mathbb{E} \left( \int_{\mathcal{X}} \frac{1}{2} \frac{\delta \lambda(x)^2}{\bar{\lambda}(x)^2} d\mathbb{N}(x) \right) \\
&= \mathbb{E} \left( \int_{\mathcal{X}} \frac{1}{2} \frac{\delta \lambda(x)^2}{\bar{\lambda}(x)^2} \lambda[k^*](x) dx \right) \\
&\leq \frac{c_2}{2c_1^2} \mathbb{E} \left( \int_{\mathcal{X}} \delta \lambda(x)^2 dx \right).
\end{aligned}
$$

For each trajectory $\mathcal{H}_T \in \mathcal{E}$, the probability density of $\mathcal{H}_T$

$$\rho(\mathcal{H}_T) = \exp \left( \int_{\mathcal{X}} \log \lambda[k^*](x) d\mathbb{N}(x) - \int_{\mathcal{X}} \lambda[k^*](x) dx \right) \leq \exp(|\mathcal{H}_T| \log c_2 - |\mathcal{M}|T c_1) =: \bar{\rho}(\mathcal{H}_T).$$

Following similar arguments as the proof of Lemma E.1,

$$\mathbb{E}\left(\ell[k^*] - \ell[k]\right) \leq \frac{c_2}{2c_1^2}\mathbb{E}\left(\int_{\mathcal{X}}\delta\lambda(x)^2 dx\right)$$

$$\leq \frac{c_2}{2c_1^2}\sum_{i=0}^{\infty}\int_{\mathcal{E}_i}\left(\int_{\mathcal{X}}\delta\lambda(x)^2 dx\right)\bar{\rho}(\mathcal{H}_T)dx_1\cdots dx_i$$

$$= \frac{c_2}{2c_1^2}\sum_{i=2}^{\infty}\frac{1}{(i-2)!}\int_{\mathcal{X}}|\mathcal{M}|^{i-2}T^{i-2}c_2^i\exp(-c_1|\mathcal{M}|T)\left(\int_{\mathcal{X}_{t(x)}}\delta k(x',x)dx'\right)^2 dx$$

$$+ \frac{c_2}{2c_1^2}\int_{\mathcal{X}}\int_{\mathcal{X}_{t(x)}}\delta k(x',x)^2\sum_{i=1}^{\infty}\frac{1}{(i-1)!}|\mathcal{M}|^{i-1}T^{i-1}c_2^i\exp(-c_1|\mathcal{M}|T)dx'dx$$

$$= \frac{c_2}{2c_1^2}\int_{\mathcal{X}}c_2^2\exp((c_2-c_1)|\mathcal{M}|T)\left(\int_{\mathcal{X}_{t(x)}}\delta k(x',x)dx'\right)^2 dx$$

$$+ \frac{c_2}{2c_1^2}\int_{\mathcal{X}}\int_{\mathcal{X}_{t(x)}}\delta k(x',x)^2 c_2\exp((c_2-c_1)|\mathcal{M}|T)dx'dx.$$

By Cauchy-Schwarz inequality,

$$\frac{c_2}{2c_1^2}\int_{\mathcal{X}}c_2^2\exp((c_2-c_1)|\mathcal{M}|T)\left(\int_{\mathcal{X}_{t(x)}}\delta k(x',x)dx'\right)^2 dx$$

$$\leq \frac{c_2}{2c_1^2}\int_{\mathcal{X}}c_2^2\exp((c_2-c_1)|\mathcal{M}|T)|\mathcal{X}_{t(x)}|\int_{\mathcal{X}_{t(x)}}\delta k(x',x)^2 dx'dx$$

$$\leq \frac{c_2}{2c_1^2}\int_{\mathcal{X}}c_2^2\exp((c_2-c_1)|\mathcal{M}|T)|\mathcal{M}|T\int_{\mathcal{X}_{t(x)}}\delta k(x',x)^2 dx'dx$$

$$= \frac{c_2^3|\mathcal{M}|T}{2c_1^2}\exp((c_2-c_1)|\mathcal{M}|T)\|\delta k\|_2^2.$$

So together we have

$$\mathbb{E}\left(\ell[k^*] - \ell[k]\right) \leq \frac{c_2^3|\mathcal{M}|T + c_2^2}{2c_1^2}\exp((c_2-c_1)|\mathcal{M}|T)\|k - k^*\|_2^2,$$

$\square$

Then we get back to the proof of Theorem D.1.

*Proof of Theorem D.1.* Let $A_0 \in \mathcal{A}_R$ be the one which minimizes $\|k^* - k_{A_0}\|_2$, i.e.,

$$\|k^* - k_{A_0}\|_2 = D(k^*, \mathcal{K}_{\text{finite}}).$$

By Lemma E.1 and Lemma E.2, there is

$$\frac{c_2^3|\mathcal{M}|T + c_2^2}{2c_1^2}\exp((c_2-c_1)|\mathcal{M}|T)D(k^*, \mathcal{K}_{\text{finite}})^2 \geq \mathbb{E}\left(\ell[k^*] - \ell[k_{A_0}]\right)$$

$$\geq \mathbb{E}\left(\ell[k^*] - \ell[k_{\widetilde{A}}]\right) \geq \frac{c_1^2}{2c_2^2}\exp(-(c_2-c_1)|\mathcal{M}|T)\|k^* - k_{\widetilde{A}}\|_2^2.$$

$\square$

### E.4 PROOF OF THEOREM D.2

Let $\overline{\mathcal{A}} = \{A \in \mathbb{R}^{S\times S}, k[A] \in \overline{\mathcal{K}}\}$. We prove the theorem under the following assumption:

**Assumption E.1.** *Assume $\widetilde{A}$ is the unique minimizer in (11), is of rank exactly $R$, and $\mathrm{vec}(\widetilde{A})$ is on the interior of $\mathrm{vec}(\overline{\mathcal{A}}) := \{\mathrm{vec}(A) : A \in \overline{\mathcal{A}}\} \subseteq \mathbb{R}^{S^2}$.*

*Proof.* Consider the local parametrization of $\mathcal{A}_R$ in the neighborhood of $\widetilde{A}$ based on the singular value decomposition $\widetilde{A} = U\Lambda V^T, U, V \in \mathbb{R}^{S \times R}, \Lambda$ is a $R$-by-$R$ diagonal matrix,

$$A = (U \; \overline{U}) \begin{pmatrix} \Lambda + P_1 & P_2^T \\ P_3 & P_3(\Lambda + P_1)^{-1}P_2^T \end{pmatrix} \begin{pmatrix} V^T \\ \overline{V}^T \end{pmatrix}.$$

Here $(U \; \overline{U}), (V \; \overline{V})$ are fixed orthogonal matrices where the first $R$ columns are $U, V$ respectively. $P_1 \in \mathbb{R}_{R \times R}, P_2, P_3 \in \mathbb{R}^{(S-R) \times R}$ are parameters in the neighborhood of 0 such that $\Lambda + P_1$ is non-singular. By Theorem 3.2 of White (1982), let $\widehat{P}_{1,\mathrm{MLE}}, \widehat{P}_{2,\mathrm{MLE}}, \widehat{P}_{3,\mathrm{MLE}}$ be the parameters which correspond to $\widehat{A}_{\mathrm{MLE}}$, i.e.

$$\widehat{A}_{\mathrm{MLE}} = (U \; \overline{U}) \begin{pmatrix} \Lambda + \widehat{P}_{1,\mathrm{MLE}} & \widehat{P}_{2,\mathrm{MLE}}^T \\ \widehat{P}_{3,\mathrm{MLE}} & \widehat{P}_{3,\mathrm{MLE}}(\Lambda + \widehat{P}_{1,\mathrm{MLE}})^{-1}\widehat{P}_{2,\mathrm{MLE}}^T \end{pmatrix} \begin{pmatrix} V^T \\ \overline{V}^T \end{pmatrix}.$$

For simplicity, let $p = (\mathrm{vec}^T(P_1), \mathrm{vec}^T(P_2), \mathrm{vec}^T(P_3))^T$. Then

$$\sqrt{M}\widehat{p}_{\mathrm{MLE}} = \sqrt{M} \begin{pmatrix} \mathrm{vec}(\widehat{P}_{1,\mathrm{MLE}}) \\ \mathrm{vec}(\widehat{P}_{2,\mathrm{MLE}}) \\ \mathrm{vec}(\widehat{P}_{3,\mathrm{MLE}}) \end{pmatrix} \xrightarrow{D} \mathcal{N}(0, I^{-1}JI^{-1}),$$

where

$$I = \mathbb{E}\left(\frac{\partial^2 \ell_A}{\partial p \partial p^T}\right)\bigg|_{P_1=0, P_2=0, P_3=0},$$

$$J = \mathbb{E}\left(\frac{\partial \ell_A}{\partial p}\frac{\partial \ell_A}{\partial p^T}\right)\bigg|_{P_1=0, P_2=0, P_3=0}.$$

For simplicity, from now on, we write all partial derivatives taken at $P_1 = 0, P_2 = 0, P_3 = 0, A = \widetilde{A}$ without specifying the location. Let

$$\gamma = \mathrm{vec}(A - \widetilde{A}) - \frac{\partial \mathrm{vec}(A)}{\partial p^T}p,$$

since $\|\gamma\| = O(\|p\|_2^2)$, there is

$$\sqrt{M}\gamma \xrightarrow{P} 0.$$

$$\sqrt{M}\mathrm{vec}(\widehat{A}_{\mathrm{MLE}} - \widetilde{A}) = \sqrt{M}\left(\gamma + \frac{\partial \mathrm{vec}(A)}{\partial p^T}\widehat{p}_{\mathrm{MLE}}\right) \xrightarrow{D} \mathcal{N}\left(0, \frac{\partial \mathrm{vec}(A)}{\partial p^T}I^{-1}JI^{-1}\frac{\partial \mathrm{vec}^T(A)}{\partial p}\right).$$

Next we look at the covariance matrix of the multivariate Gaussian distribution,

$$\frac{\partial \ell_A}{\partial p} = \frac{\partial \mathrm{vec}^T(A)}{\partial p}\frac{\partial \ell_A}{\partial \mathrm{vec}(A)}.$$

Since $\widetilde{A}$ maximizes $\mathbb{E}(\ell_A)$ and is on the interior of $\overline{\mathcal{A}}$ and hence the interior of the manifold parametrized by $p$, we have

$$\mathbb{E}\left(\frac{\partial \ell_A}{\partial p}\right) = \frac{\partial \mathrm{vec}^T(A)}{\partial p}\mathbb{E}\left(\frac{\partial \ell_A}{\partial \mathrm{vec}(A)}\right) = \frac{\partial \mathrm{vec}^T(A)}{\partial p}\mathrm{vec}(\widetilde{G}) = 0. \tag{18}$$

$$\begin{aligned} J &= \mathbb{E}\left(\frac{\partial \ell_A}{\partial p}\frac{\partial \ell_A}{\partial p^T}\right) \\ &= \frac{\partial \mathrm{vec}^T(A)}{\partial p}\mathbb{E}\left(\frac{\partial \ell_A}{\partial \mathrm{vec}(A)}\frac{\partial \ell_A}{\partial \mathrm{vec}^T(A)}\right)\frac{\partial \mathrm{vec}(A)}{\partial p^T} \\ &= \frac{\partial \mathrm{vec}^T(A)}{\partial p}\left(\mathrm{Cov}\left(\frac{\partial \ell_A}{\partial \mathrm{vec}(A)}, \frac{\partial \ell_A}{\partial \mathrm{vec}^T(A)}\right) + \mathbb{E}\left(\frac{\partial \ell_A}{\partial \mathrm{vec}(A)}\right)\mathbb{E}\left(\frac{\partial \ell_A}{\partial \mathrm{vec}(A)}\right)^T\right)\frac{\partial \mathrm{vec}(A)}{\partial p^T} \\ &= \frac{\partial \mathrm{vec}^T(A)}{\partial p}\widetilde{J}\frac{\partial \mathrm{vec}(A)}{\partial p^T}. \end{aligned}$$

For any $i, j \in [(2S - R)R]$,

$$\frac{\partial^2 \ell_A}{\partial p_i \partial p_j} = \frac{\partial \text{vec}^T(A)}{\partial p_i} \frac{\partial^2 \ell_A}{\partial \text{vec}(A) \partial p_j} + \frac{\partial \text{vec}^T(A)}{\partial p_j \partial p_j} \frac{\partial \ell_A}{\partial \text{vec}(A)}$$

$$= \frac{\partial \text{vec}^T(A)}{\partial p_i} \frac{\partial^2 \ell_A}{\partial \text{vec}(A) \partial \text{vec}^T(A)} \frac{\partial \text{vec}(A)}{\partial p_j} + \frac{\partial^2 \text{vec}^T(A)}{\partial p_j \partial p_j} \frac{\partial \ell_A}{\partial \text{vec}(A)},$$

the (i,j)-th component of $I$

$$I_{ij} = \mathbb{E}\left(\frac{\partial^2 \ell_A}{\partial p_i \partial p_j}\right) = \frac{\partial \text{vec}^T(A)}{\partial p_i} \widetilde{I} \frac{\partial \text{vec}(A)}{\partial p_j} + \frac{\partial^2 \text{vec}^T(A)}{\partial p_j \partial p_j} \text{vec}(\widetilde{G}).$$

We observe that some blocks of $(\frac{\partial^2 \text{vec}^T(A)}{\partial p_j \partial p_j} \text{vec}(\widetilde{G}))_{ij}$ are 0.

$$\frac{\partial^2 \text{vec}^T(A) \text{vec}(\widetilde{G})}{\partial \text{vec}(P_1) \partial \text{vec}^T(P_1)} = \frac{\partial^2 \text{vec}^T(UP_1V^T + UP_2^T\overline{V}^T + \overline{U}P_3V^T + \overline{U}P_3(\Lambda + P_1)^{-1}P_2^T\overline{V}^T) \text{vec}(\widetilde{G})}{\partial \text{vec}(P_1) \partial \text{vec}^T(P_1)} = 0,$$

as (1) the second order derivative of $UP_1V^T, UP_2^T\overline{V}^T, \overline{U}P_3V^T$ over $P_1$ is 0 since they are either linear in or irrelevant with $P_1$, (2) the second order derivative of $UP_1V^T + \overline{U}P_3(\Lambda + P_1)^{-1}P_2^T\overline{V}^T$ over $P_1$ is 0 since it is taken at $P_2 = P_3 = 0$. Also we have

$$\frac{\partial^2 \text{vec}^T(A) \text{vec}(\widetilde{G})}{\partial \text{vec}(P_1) \partial \text{vec}^T(P_2)} = 0, \quad \frac{\partial^2 \text{vec}^T(A) \text{vec}(\widetilde{G})}{\partial \text{vec}(P_1) \partial \text{vec}^T(P_3)} = 0,$$

$$\frac{\partial^2 \text{vec}^T(A) \text{vec}(\widetilde{G})}{\partial \text{vec}(P_2) \partial \text{vec}^T(P_2)} = 0, \quad \frac{\partial^2 \text{vec}^T(A) \text{vec}(\widetilde{G})}{\partial \text{vec}(P_3) \partial \text{vec}^T(P_3)} = 0,$$

for similar reasons. The only non-zero block is

$$\frac{\partial^2 \text{vec}^T(A) \text{vec}(\widetilde{G})}{\partial \text{vec}(P_2) \partial \text{vec}^T(P_3)} = \frac{\partial^2 \text{vec}^T(\overline{U}P_3\Lambda^{-1}P_2^T\overline{V}^T) \text{vec}(\widetilde{G})}{\partial \text{vec}(P_2) \partial \text{vec}^T(P_3)}$$

$$= \frac{\partial}{\partial \text{vec}(P_2)} \frac{\partial \text{vec}^T(\overline{U}P_3\Lambda^{-1}P_2^T\overline{V}^T) \text{vec}(\widetilde{G})}{\partial \text{vec}^T(P_3)}$$

$$= \frac{\partial}{\partial \text{vec}(P_2)} \frac{\partial \text{vec}^T(P_3)(\Lambda^{-1}P_2^T\overline{V}^T \otimes \overline{U}^T) \text{vec}(\widetilde{G})}{\partial \text{vec}^T(P_3)}$$

$$= \frac{\partial}{\partial \text{vec}(P_2)} ((\Lambda^{-1}P_2^T\overline{V}^T \otimes \overline{U}^T) \text{vec}(\widetilde{G}))^T$$

$$= \frac{\partial}{\partial \text{vec}(P_2)} \text{vec}^T(\overline{U}^T \widetilde{G} \overline{V} P_2 \Lambda^{-1})$$

$$= \frac{\partial}{\partial \text{vec}(P_2)} \text{vec}^T(P_2)(\Lambda^{-1} \otimes \overline{V}^T \widetilde{G}^T \overline{U})$$

$$= \Lambda^{-1} \otimes \overline{V}^T \widetilde{G}^T \overline{U}.$$

So

$$I = \frac{\partial \text{vec}^T(A)}{\partial p} \widetilde{I} \frac{\partial \text{vec}(A)}{\partial p^T} + \begin{pmatrix} 0 & 0 & 0 \\ 0 & 0 & \Lambda^{-1} \otimes \overline{V}^T \widetilde{G}^T \overline{U} \\ 0 & \Lambda^{-1} \otimes \overline{U}^T \widetilde{G} \overline{V} & 0 \end{pmatrix}.$$

At last, we compute $\partial \text{vec}(A)/\partial p^T$,

$$\frac{\partial \text{vec}(A)}{\partial p^T} = \begin{pmatrix} \dfrac{\partial \text{vec}(A)}{\partial \text{vec}^T(P_1)} & \dfrac{\partial \text{vec}(A)}{\partial \text{vec}^T(P_2)} & \dfrac{\partial \text{vec}(A)}{\partial \text{vec}^T(P_3)} \end{pmatrix}$$

$$= \begin{pmatrix} \dfrac{\partial \text{vec}(UP_1V^T)}{\partial \text{vec}^T(P_1)} & \dfrac{\partial \text{vec}(UP_2^T\overline{V}^T)}{\partial \text{vec}^T(P_2)} & \dfrac{\partial \text{vec}(\overline{U}P_3V^T)}{\partial \text{vec}^T(P_3)} \end{pmatrix}$$

$$= \begin{pmatrix} V \otimes U & (\overline{V} \otimes U)Q_{S-R,R} & V \otimes \overline{U} \end{pmatrix}.$$

Let

$$
Z = \begin{pmatrix} \mathbb{I}_{R^2} & & \\ & Q_{R,S-R} & \\ & & \mathbb{I}_{R(S-R)} \end{pmatrix} \begin{pmatrix} V^T \otimes \mathbb{I}_R & \mathbb{I}_R \otimes U^T \\ \overline{V}^T \otimes \mathbb{I}_R & \\ & \mathbb{I}_R \otimes \overline{U}^T \end{pmatrix},
$$

$Z$ has full row-rank, and

$$
\frac{\partial \text{vec}(A)}{\partial p^T} Z = \begin{pmatrix} V \otimes U & \overline{V} \otimes U & V \otimes \overline{U} \end{pmatrix} \begin{pmatrix} V^T \otimes \mathbb{I}_R & \mathbb{I}_R \otimes U^T \\ \overline{V}^T \otimes \mathbb{I}_R & \\ & \mathbb{I}_R \otimes \overline{U}^T \end{pmatrix}
$$

$$
= \begin{pmatrix} (VV^T + \overline{V}\overline{V}^T) \otimes U & V \otimes (UU^T + \overline{U}\overline{U}^T) \end{pmatrix}
$$

$$
= F.
$$

Then the covariance matrix of the asymptotic distribution of $\text{vec}(\widehat{A}_{\text{MLE}})$

$$
\frac{\partial \text{vec}(A)}{\partial p^T} I^{-1} J I^{-1} \frac{\partial \text{vec}^T(A)}{\partial p} = \frac{\partial \text{vec}(A)}{\partial p^T} Z (Z^T I Z)^{\dagger} Z^T J Z (Z^T I Z)^{\dagger} Z^T \frac{\partial \text{vec}^T(A)}{\partial p}
$$

$$
= F(Z^T I Z)^{\dagger} F^T \widetilde{J} F (Z^T I Z)^{\dagger} F^T.
$$

Next we check that

$$
(Z^T I Z) = F^T (\widetilde{I} + \widetilde{C}) F.
$$

By (18), we know

$$
\begin{pmatrix} \text{vec}^T(U^T \widetilde{G}) & \text{vec}^T(\widetilde{G}V) \end{pmatrix} = \text{vec}^T(\widetilde{G}) F = \text{vec}^T(\widetilde{G}) \frac{\partial \text{vec}(A)}{\partial p^T} Z = 0.
$$

So

$$
\overline{U}\overline{U}^T \widetilde{G} = (\mathbb{I}_S - UU^T)\widetilde{G} = \widetilde{G} = \widetilde{G}(\mathbb{I}_S - VV^T) = \widetilde{G}\overline{V}\overline{V}^T.
$$

$$
Z^T I Z - F^T \widetilde{I} F
$$

$$
= Z^T \begin{pmatrix} 0 & 0 & 0 \\ 0 & 0 & \Lambda^{-1} \otimes \overline{V}^T \widetilde{G}^T \overline{U} \\ 0 & \Lambda^{-1} \otimes \overline{U}^T \widetilde{G}\overline{V} & 0 \end{pmatrix} Z
$$

$$
= \begin{pmatrix} V^T \otimes \mathbb{I}_R & \mathbb{I}_R \otimes U^T \\ \overline{V}^T \otimes \mathbb{I}_R & \\ & \mathbb{I}_R \otimes \overline{U}^T \end{pmatrix}^T \begin{pmatrix} 0 & 0 & 0 \\ 0 & 0 & Q_{S-R,R}(\Lambda^{-1} \otimes \overline{V}^T \widetilde{G}^T \overline{U}) \\ 0 & (\Lambda^{-1} \otimes \overline{U}^T \widetilde{G}\overline{V})Q_{R,S-R} & 0 \end{pmatrix}
$$

$$
\begin{pmatrix} V^T \otimes \mathbb{I}_R & \mathbb{I}_R \otimes U^T \\ \overline{V}^T \otimes \mathbb{I}_R & \\ & \mathbb{I}_R \otimes \overline{U}^T \end{pmatrix}
$$

$$
= \begin{pmatrix} 0 & ((\mathbb{I}_R \otimes \overline{U})(\Lambda^{-1} \otimes \overline{U}^T \widetilde{G}\overline{V})Q_{R,S-R}(\overline{V}^T \otimes \mathbb{I}_R))^T \\ (\mathbb{I}_R \otimes \overline{U})(\Lambda^{-1} \otimes \overline{U}^T \widetilde{G}\overline{V})Q_{R,S-R}(\overline{V}^T \otimes \mathbb{I}_R) & 0 \end{pmatrix}
$$

$$
= \begin{pmatrix} 0 & ((\mathbb{I}_R \Lambda^{-1} \mathbb{I}_R \otimes \overline{U}\overline{U}^T \widetilde{G}\overline{V}\overline{V}^T)Q_{R,S})^T \\ (\mathbb{I}_R \Lambda^{-1} \mathbb{I}_R \otimes \overline{U}\overline{U}^T \widetilde{G}\overline{V}\overline{V}^T)Q_{R,S} & 0 \end{pmatrix}
$$

$$
= \begin{pmatrix} 0 & ((\Lambda^{-1} \otimes \widetilde{G})Q_{R,S})^T \\ (\Lambda^{-1} \otimes \widetilde{G})Q_{R,S} & 0 \end{pmatrix}.
$$

$$\begin{aligned}
F^T(V\Lambda^{-1}U^T \otimes \widetilde{G})Q_{S,S}F &= \begin{pmatrix} \mathbb{I}_S \otimes U^T \\ V^T \otimes \mathbb{I}_S \end{pmatrix}(V\Lambda^{-1}U^T \otimes \widetilde{G})Q_{S,S}\left(\mathbb{I}_S \otimes U \; V \otimes \mathbb{I}_S\right) \\
&= \begin{pmatrix} V\Lambda^{-1}U^T \otimes 0 \\ \Lambda^{-1}U^T \otimes \widetilde{G} \end{pmatrix}Q_{S,S}\left(\mathbb{I}_S \otimes U \; V \otimes \mathbb{I}_S\right) \\
&= \begin{pmatrix} 0 \\ Q_{R,S}\widetilde{G} \otimes \Lambda^{-1}U^T \end{pmatrix}\left(\mathbb{I}_S \otimes U \; V \otimes \mathbb{I}_S\right) \\
&= \begin{pmatrix} 0 & 0 \\ Q_{R,S}(\widetilde{G} \otimes \Lambda^{-1}) & 0 \end{pmatrix} \\
&= \begin{pmatrix} 0 & 0 \\ (\Lambda^{-1} \otimes \widetilde{G})Q_{R,S} & 0 \end{pmatrix},
\end{aligned}$$

$$F^T\widetilde{C}F = \begin{pmatrix} 0 & 0 \\ (\Lambda^{-1} \otimes \widetilde{G})Q_{R,S} & 0 \end{pmatrix} + \begin{pmatrix} 0 & 0 \\ (\Lambda^{-1} \otimes \widetilde{G})Q_{R,S} & 0 \end{pmatrix}^T = Z^TIZ - F^T\widetilde{I}F.$$

$\square$

