# OpenReview forum: "Neural Spectral Marked Point Processes"
_ICLR.cc/2022/Conference — ICLR 2022 Poster_

### Official Review · Reviewer_q7Hf · 2021-10-30

**Correctness:** 3
**Technical Novelty And Significance:** 2
**Empirical Novelty And Significance:** 2
**Recommendation:** 3
**Confidence:** 3

**Main Review:**

The paper is correctly written and claims are well supported by references. However, mathematical proofs are not correctly written, the best example si page 17 where authors stack equations with several unnecessary informations leaving the reader to a rough experience.


Experimental part is based on mostly synthetic data set in low dimension and two real-world data set is not enough to show advantages of your approach. Further, it is clearly incomplete:

- What about the possibility of modeling $\phi$ and $\psi$ ? You should study several architectures, parameters ... in order to explore your approach.
- What about computation time ? it seems that NMPP have computational cost much higher than classical Hawkes process where kernel have a closed-form that does not rely on NNs.
- Why using SGD in order to optimize the MLE rather than widely used adam or rmsprop or others...?


**Summary Of The Paper:**

The paper propose to replace the kernel involving in Marked Point Processes (MPP) by the product of two neural networks.

**Summary Of The Review:**

Theoretical results appears to have very few links with empirical results and the behavior of the proposed approach in practice. Furthermore, the experimental part is not properly studied and does not support the advantages of their approach. Limitations of the proposed approach are not investigated. The paper could be a good paper with more complete work.

---

> ### Author Response · Authors · 2021-11-21
> **Response to Reviewer q7Hf**
>
> We thank the reviewer's comments. We clarified in the original version that the estimation of the influence kernel function for the point process is essential because it is the most important component in the model and offers clear interpretations for practice. Moreover, the theoretical results has proven that the true kernel function is identifiable by solving the MLE problem under some regularity conditions which is of great importance when learning the kernel by neural networks. Thus it provides guarantees for our learned kernel to be smooth and interpretable in practice. The limitations of our approach were discussed in section 5 in the original version.
>
> (1)  We have improved the presentation of the proof by breaking down the multi-line equations into several pieces in Appendix C (current page 17, 18, and 19), with more explanations to clarify the steps. The proof itself is the same as before, and we believe it is mathematically correct.
>
> (2) We have added additional experiments to demonstrate that the results are not sensitive to the choice of network architecture and hyper-parameters, by increasing the depth and trying different architectures in Appendix E. With varying architectures, all models succeed in recovering the true kernel. Moreover, in Appendix B we studied that when the basis representation of $\phi$ and $\psi$ are random features, the model can be naturally viewed as a neural network with one hidden layer. Thus our proposed method acts as a fundamental framework of finite-rank and neural network kernel representation. Generally, model architectures may well depend on specific tasks. For example, neural networks can be based on CNN if the high dimensional markers are image-like and can also be LSTM or even BERT if the markers are text-based. The choice of deep model architecture and optimization algorithm can be more systematically explored, which we comment in the Discussion Section.
>
> (3) The performance comparison has been shown in the response to reviewer 2 (Table 1 *Running time*). The training time of our model is similar with other neural point process models (RMTPP and NH). The classcial Hawkes model (Hawkes) has the minimum training time because it has only two parameters to be estimated.
>
> (4) The neural network training in all reported experiments are actually conducted using Adam -- which we referred to as SGD since Adam is a special variant of SGD. We have added ``experimental detail'' in  Appendix D to clarify this. Other SGD methods like RMSprop can also be explored. We use Adam due to its better empirical performance.

---

### Official Review · Reviewer_HJXe · 2021-10-31

**Correctness:** 4
**Technical Novelty And Significance:** 3
**Empirical Novelty And Significance:** Not applicable
**Recommendation:** 8
**Confidence:** 3

**Main Review:**

Strengths:
- Novel method to model complex dependencies in temporal and spatio-temporal event data using the proposed non-stationary kernel, which experimental results shows the superior performance of the method.
- Theoretical guarantees to find the kernels using MLE (although not fully check the proof).

Weaknesses:
- In contrast to synthetic data the real data experiment is very brief. Authors compared the proposed method using only likelihood, but it is better to also include some methods to measure the predictive power of various methods to generate events.
- Unfortunately, there is no details about the structure and size of the neural networks which used to model the kernel functions. Also how the performance of the method is related to the number of parameters of NNs and number of events data that used for training.
- It is better to include the comparison of the proposed method in terms on training and test/inference time.


**Summary Of The Paper:**

This paper propose a point process with a non-stationary kernel to model complex event data. The kernel represented by its finite rank decomposition and the basis functions (feature functions) are models using a neural network architecture. To learn the model parameters they used stochastic gradient to maximize the resulting likelihood function. Moreover, they give a theoretical guarantee by showing that under some assumptions, true kernel function is identifiable by solving the MLE problem. They also compared their method against state-of-the-art and baseline on synthetic and real datasets.

**Summary Of The Review:**

In overall the proposed method for modeling the event data is novel enough and seems to have superior performance in comparison to other state-of-the-arts methods. It also can capture more complex behaviors in events data.

---

> ### Author Response · Authors · 2021-11-21
> **Response to Reviewer HJXe**
>
> We thank the reviewer's positive comments.
>
> (1) We also want to clarify that because the ground truth for real data is unavailable and event generation in point process is stochastic which cannot be measured exactly, it is reasonable and reliable for people to demonstrate the goodness of model using likelihood. We also use out-of-sample predictive log-likelihood to demonstrate the predictive power of our model.
>
> (2)  We thank the reviewer's comment. Please note that in the original submission, the hyper-parameters and details about the neural network architecture were presented in Appendix D, as follows. We set rank $R=5$ in our setting. We consider the shared network that summarizes input data into a hidden embedding to be a fully connected three-layer network and the sub-network to be a fully connected network with two hidden layers. The width of the hidden layers in the shared network and sub-network is $n=128$, and the width of the input layers in sub-networks (or the output layer in the shared network) is $p=10$. We adopt the SoftPlus as the activation function of each layer in the network.
>
> We also add additional experiments to demonstrate our proposed method's robust performance with different neural network architectures (numbers of parameters) and different training sample sizes. The results are shown in Figures 8 and 9 in Appendix E. In Figure 8, the data is generated using exponential decaying stationary kernel and Figure 9 uses the same data set as in Figure 3. The *Proposed kernel* refers to the model with the original experiment settings (in Figure 3). The *Proposed kernel with increased network size* refers to the model with one more hidden layer and doubled layer width in the sub-networks ($p=20$). The *Proposed kernel with half training sample size* represents the model with a default network architecture but only trained with half of the training samples. The results show that our proposed model works consistently well without overfitting after increasing the model size or decreasing the training sample size in both the stationary and non-stationary cases. This also demonstrates that the NSMPP model is adaptive: while having the capacity to express the complicated non-stationary kernel, it can also approximate the standard parametric stationary kernel without incurring overfitting.
>
> (3) In the revised paper, we have compared the running time of our model and other baselines: Recurrent marked temporal point processes (RMTPP), neural Hawkes process (NH), and standard Hawkes process (Hawkes) per batch on 1D synthetic data set 4. We adopt Adam optimizer with a constant learning rate $10^{-2}$ and the batch size is 25. The experiments are performed on Google Colaboratory (Pro version) with 25GB RAM and a Tesla P100 GPU. The size and performance of each model are presented in Table 1 \textit{Running time}. The running time refers to wall clock time. <br> The training time of our model is similar with other neural point process models (RMTPP and NH). The classical Hawkes model (Hawkes) has the minimum training time because it has only two parameters to be estimated. The testing/inference time of all models to compute the conditional intensity function are less than one seconds.
>
> Table1. Running time
>
> | | NSMPP | RMTPP | NH | Hawkes |
> |------|------|------|------|------|
> | Number of parameters  | 171555 | 952384 | 198406 | 2 |
> | Running Time (seconds/per batch)  | 0.766 | 0.834 | 0.682 | 0.113 |
>
> The training time of our model is similar with other neural point process models (RMTPP and NH). The classical Hawkes model (Hawkes) has the minimum training time because it has only two parameters to be estimated. The testing/inference time of all models to compute the conditional intensity function are less than one seconds.

---

### Official Review · Reviewer_CU9d · 2021-11-08

**Correctness:** 3
**Technical Novelty And Significance:** 2
**Empirical Novelty And Significance:** 2
**Recommendation:** 6
**Confidence:** 4

**Main Review:**


I find that this paper has interesting ideas, and it addresses an important question in learning self-exciting processes in the presence of non-stationarity.

However, I think the authors missed an extremely important influential paper by Feng Chen and Peter Hall (2013): https://www.cambridge.org/core/journals/journal-of-applied-probability/article/inference-for-a-nonstationary-selfexciting-point-process-with-an-application-in-ultrahigh-frequency-financial-data-modeling/99E7C55841B746DBD6CE20D657B2CC9D

Of course, the modulation of the excitation function are very different, where the authors used a general function for k(\cdot, \cdot). In Chen/Hall's paper, they had the remarkable idea that it MLE estimates be accurately inferred when you have many paths but with T (time to horizon fixed) and not otherwise. In their paper they set T = 1. I would like to see how your framework would subsume theirs (in the most vanilla case), as your k(\cdot,\cdot) is that of a much general form.



**Summary Of The Paper:**

In this paper, the authors introduce a neural network framework modulating the excitation functions within the self and mutually exciting count processes.

**Summary Of The Review:**

This is an important piece of work, but I would like to see how they contrast their work with Chen/Hall, which I think they missed.

---

> ### Author Response · Authors · 2021-11-21
> **Response to Reviewer CU9d**
>
>  We thank the reviewer’s positive comments and for bringing up the paper by Chen/Hall. The citation and comparison are included in the updated version of the manuscript. While the paper also considers modeling non-stationarity self-exciting point processes, Chen/Hall considers a different type of non-stationary self-exciting processes: Chen/Hall considers time-varying background intensity of the point process, while we focus on the non-stationary triggering kernel -- which focuses on the self-exciting triggering part.

---

### Author Response · Authors · 2021-11-23
**Thank reviewers' valuable comments; please see author response.**

We thank the reviewers for the valuable feedback and constructive comments. In below, we address the questions to each reviewer. We have also revised the manuscript based on the comments, where the major revisions are marked in blue.

---

### Decision · Program_Chairs · 2022-01-20

**Decision:**

Accept (Poster)

**Comment:**

This paper proposes a self-exciting temporal point process model with a non-stationary triggering kernel to model complex dependencies in temporal and spatio-temporal event data. The kernel is represented by its finite rank decomposition and a set of neural basis functions (feature functions). The proposed model has superior performance in comparison to other state-of-the-arts methods. All the reviewers recognized that the model is interesting and advances the state of the art in a meaningful way. While they were some concerns regarding the experimental evaluation, particularly in terms of real data, and the presentation, the rebuttal/revision by the authors cleared up these concerns.